

# A National Scale Hybrid Model for Enhanced Streamflow Estimation - Consolidating a Physically Based Hydrological Model with Long Short-term Memory Networks

Jun Liu, Julian Koch, Simon Stisen, Lars Troldborg, Raphael J. M. Schneider

Department of hydrology, Geological Survey of Denmark and Greenland, Copenhagen, 1350, Denmark

*Correspondence to*: Jun Liu (juliu@geus.dk)

**Abstract.** Accurate streamflow estimation is essential for effective water resources management and adapting to extreme events in the face of changing climate conditions. Hydrological models have been the conventional approach for streamflow inter/extrapolation in time and space for the past decades. However, their large-scale applications have encountered challenges, 10  including issues related to efficiency, complex parameterization, and constrained performance. Deep learning methods, such as Long Short-Term Memory networks (LSTM), have emerged as a promising and efficient approach for large-scale streamflow estimation. In this study, we conducted a series of experiments to identify optimal hybrid modelling schemes to consolidate physically based models with LSTM aimed at enhancing streamflow estimation in Denmark.

The results showed that the hybrid modelling schemes outperformed the Danish Water Resources Model (DKM) in both 15  gauged and ungauged basins. While the standalone LSTM rainfall-runoff model outperformed DKM in many basins, it faced challenges when predicting streamflow in groundwater-dependent catchments. A serial hybrid modelling scheme (LSTM-q), which used DKM outputs and climate forcings as dynamic inputs for LSTM training, demonstrated higher performance. LSTM-q improved the median Nash-Sutcliffe Efficiency (NSE) by 0.18 in gauged basins and 0.11 in ungauged basins compared to DKM. Similar accuracy improvements were achieved with alternative hybrid schemes, i.e., by predicting the 20  residuals between DKM-simulated streamflow and observations using a LSTM. Moreover, the developed hybrid models enhanced the accuracy of extreme events, which encourages the integration of hybrid models within an operational forecasting framework. This study highlights the advantages of synergizing existing physically based hydrological models with LSTM models, and the proposed hybrid schemes hold the potential to achieve high-quality, large-scale streamflow estimations.

## 1 Introduction

Accurate streamflow estimates are essential for sustainable water resource management, prediction of extreme events, energy production, decision making, and the protection of both human populations and natural ecosystems (Devitt et al., 2023; Hoy, 2017; Satoh et al., 2022). Collecting spatiotemporally adequate streamflow data through observations can be challenging. Therefore, various conceptual and process-based hydrological models have been developed and applied for streamflow extra/interpolation in time and space (Beven, 1996, 2020; Devia et al., 2015). These models are based on a priori knowledge



and physical principles to simulate critical hydrological processes, e.g., infiltration, evapotranspiration, runoff routing, and groundwater movement, and have been widely and successfully used across domains and scales.

Physically based distributed models (PBMs) stand out among those diverse hydrological models and have been widely used in recent decades due to their sophisticated structures and advanced parameterizations (Devia et al., 2015; Fatichi et al., 2016; Pakoksung and Takagi, 2021; Refsgaard et al., 2022). These features enable PBMs to simulate complex hydrological processes
and facilitate detailed analysis at high spatiotemporal resolutions. However, PBMs are susceptible to biases arising from inadequate inputs, suboptimal structural design, or improper parameterization schemes (Herrera et al., 2022; Dembélé et al., 2020; Silvestro et al., 2015; Koch et al., 2016). Therefore, the streamflow performance of PBMs is not always satisfactory for practical applications and may not consistently outperform simpler lumped and conceptual hydrological models. For example, some studies have pointed out that PBMs encounter difficulties in capturing peak flows (Baroni et al., 2019; Kumari et al.,
2021; Moges et al., 2021; Sahraei et al., 2020).

The Danish Water Resources Model (DKM) is an example of PBM (Højberg et al., 2009), which is based on the distributed, integrated model code MIKE SHE (DHI, 2020). The DKM has been calibrated against a large dataset of groundwater head observations, and streamflow measurements utilizing dense national monitoring networks (Henriksen et al., 2021; Stisen et al., 2020). Streamflow performance is considered satisfactory, with an average Kling-Gupta Efficiency (KGE) of 0.75, though
performance varies both temporally and spatially. Overall, the DKM tends to exhibit better performance in basins with larger drainage areas compared to smaller ones (Henriksen et al., 2021). In recent years, several projects related to hydrological monitoring, national flood warning, and nitrate modelling have emerged that rely on DKM-simulated streamflow time series (Henriksen et al., 2023). Therefore, enhancing the accuracy of DKM simulations using advanced methods, such as deep learning (DL) algorithms, is deemed necessary and will have far reaching implications for a range of applications.

Data-driven techniques are well suited for capturing patterns and relationships within data, without relying on prior assumptions or models (Kawaguchi et al., 2022; Ke et al., 2017; Rätsch, 2004; Wu et al., 2022). The runoff process is intricately connected to climate records and other processes in the water cycle. These relationships can be learned through data-driven methods, such as LSTM (Wi and Steinschneider, 2023; Wang et al., 2023; Kratzert et al., 2018). LSTM is a type of recurrent neural network proficient in handling time series data and has proven to effectively capture the variations and dependencies
within sequential data (Hochreiter and Schmidhuber, 1997; Greff et al., 2017). It has found successful applications in hydrology, particularly for estimating streamflow in numerous catchments, with encouraging performance (Arsenault et al., 2023; Hunt et al., 2022; Cheng et al., 2020; Zhang et al., 2022; Hashemi et al., 2022; Lees et al., 2021; Wilbrand et al., 2023; Frame et al., 2021a). Nonetheless, concerns exist regarding DL methods, such as their inherently complex internal structures (Ghorbani and Zou, 2019; Goldstein et al., 2015). While they often yield higher performance, accuracy can decrease when
attempting to transfer models from gauged basins to ungauged ones. Therefore, the integration of DL methods with PBMs and the development of hybrid systems have been recognized as a promising approach to robustly enhance streamflow predictions (Slater et al., 2023). In such hybrid modelling schemes, PBMs provide a substantial amount of sequential data containing



consolidated hydrological knowledge within the simulation domain, while deep learning algorithms have the potential to exploit multiple data types and uncover information that may be overlooked or ignored by PBMs.

A straightforward approach to develop hybrid models is to set up a serial system that uses the outputs of existing PBMs as inputs for LSTM modelling (Amendola et al., 2020; Slater et al., 2023). This approach offers several benefits. For instance, they are efficient and require fewer modifications to the existing PBMs, which may have undergone decades of development and contain valuable physical knowledge. Attempts have been made in various regions where DL methods were employed to post-process imperfect PBM simulations (Cho and Kim, 2022; Frame et al., 2021b; Konapala et al., 2020; Liu et al., 2022;

Shen et al., 2022). While earlier studies have explored different hybrid systems, there remain scientific aspects that warrant further investigation:

1. What are the optimal hybrid schemes for combining PBMs and LSTM in Denmark?

While earlier studies have explored a limited number of alternative hybrid modelling schemes, the full potential of intercomparing different hybrid modelling schemes and a systematic comparison and evaluation of the alternative approaches

remains untapped. Frame et al., (2021), Tang et al., (2023), and Liu et al., (2022) evaluated the potential benefit of PBMs outputs and climate forcings as LSTM inputs, with streamflow as the target variable for prediction. Their results indicated a significant improvement in the performance of streamflow estimation by hybrid models compared to benchmark models, i.e., the National Water Model, Global Hydrological Models and WRF-Hydro. Cho and Kim (2022) and Konapala et al., (2020) investigate the performance of a LSTM model, which predicts the residuals between WRF-Hydro simulated discharge and

observations. Koch and Schneider (2022) proposed that an LSTM model pretrained with DKM simulated discharge as the target variable, followed by fine-tuning with observed discharge, yielded superior results. These studies offer intriguing approaches to consolidate PBMs with LSTM in hybrid modelling schemes. It is imperative to evaluate these approaches to identify the optimal methods.

2. How can we expand the scope of studies on LSTM models to encompass national scales and groundwater-dependent

systems?

To date, research on LSTM models has focused on rainfall-runoff processes in gauged basins, such as the Catchment Attributes and Meteorology for Large-sample Studies (CAMELS-US) dataset (Addor et al., 2017), CAMELS-UK dataset (Coxon et al., 2020), and Global Runoff Data Centre (Tang et al., 2023). Many studies have investigated local basins with limited data coverage (Cho and Kim, 2022; Hunt et al., 2022; Liu et al., 2022). However, there is a notable absence of studies that expand

simulations to national scale, i.e., making predictions for all catchments gauged and ungauged and provide a comprehensive map of biases between DL and PBM models. In our study, Denmark, delineated in 2830 catchments, serves as the study area, potentially enriching the geographical scope of this topic.

3. What is the impact of physical processes on LSTM performance in groundwater-dependent areas, and how can we bridge the gap between LSTM and physical knowledge?

Connecting LSTM with physical knowledge is an active area of research. Investigating the influence of physical processes on LSTM performance in complex hydrological settings, such as groundwater-dependent flow regimes, is crucial. While previous



studies have explored the effects of snow melting (Frame et al., 2021b; Fuente et al., 2023; Kratzert et al., 2019a; Wang et al., 2022) on LSTM modelling, limited attention has been given to the impacts of groundwater variations on LSTM rainfall-runoff modelling. This gap may be due to the scarcity of observations or the absence of well-established groundwater modelling

systems like DKM to support such analyses. Therefore, DKM serves as a valuable testbed for investigating the enhancement of physically informed data-driven models in groundwater-dependent regions.

4. What is the potential of LSTM hybrid models for streamflow estimation in operational frameworks, especially for extreme events?

As the frequency of extreme events is projected to increase in the coming decades, there is growing demand for real-time

modelling and forecasting (Curceac et al., 2020; Devitt et al., 2023; Hauswirth et al., 2021). Operational real-time modelling and forecasting frameworks are thus under development with the primary objective of delivering timely warnings, usually based on a short simulation period of hindcasting, nowcasting and forecasting (Nevo et al., 2022). In this context, only few studies have investigated the potential applicability of LSTM hybrid schemes on short simulation periods with a focus of extreme events.   Hunt et al. (2022) examined the performance of LSTM models trained to ingest catchment-mean

meteorological and hydrological variables from the Global Flood Awareness System (GloFAS)–ERA5 reanalysis and output streamflow at ten hydrological stations in the western US. They utilized the European Centre for Medium-Range Weather Forecasts (ECMWF) Integrated Forecasting System (IFS) to feed the models, predicting streamflow with a lead time of ten days. Their study demonstrated the potential of hybrid LSTM models in the context of operational forecast.  The developed LSTM hybrid schemes from this study are expected to support the initiative towards operational modelling in Denmark. Thus,

the developed models are specifically assessed during extreme events.

The aim of this study is to test various hybrid systems combining LSTM and DKM and identify optimal LSTM hybrid schemes tailored to streamflow modelling, with applicability in generating continuous streamflow predictions across Denmark with daily timestep.

## 2 Methods and data

This section begins with a description of the LSTM algorithm (Section 2.1), followed by data collection and the introduction of the benchmark model, DKM and LSTM rainfall-runoff model (Section 2.2). Subsequently, Section 2.3 outlines various candidate LSTM hybrid modelling schemes. Details regarding the experiment designs are provided in Section 2.4, and Section 2.5 presents the description of evaluation metrics for assessing model performance.

## 2.1 Long short-term memory networks (LSTM)

LSTM is a type of recurrent neural networks (RNNs) specifically developed to address the shortcomings of traditional RNNs when confronted with sequences featuring long-term dependencies (Hochreiter and Schmidhuber, 1997; Sutskever et al., 2014; Rahmani et al., 2020; Gers et al., 2000; Greff et al., 2017; Kratzert et al., 2018). These networks possess the remarkable ability





to selectively retain or discard information over extended sequences. They achieve this by using specialized memory cells that
store and update information as it traverses the networks (Gers et al., 2000). LSTM networks are equipped with multiple hidden
unit sizes and incorporate essential information processing instants, namely the input, forget, and output gates. These gates
play main roles in regulating the flow of sequential information, enabling the network to determine what information should
be preserved and what should be discarded at each time step. While a comprehensive understanding of LSTM networks can
be found in numerous studies, readers with a background in hydrology are encouraged to explore the works of Kratzert et al.
(2018) for more detailed insights.

**2.2 Dataset and benchmark model**

**2.2.1 ID15 catchments**

For various water management tasks, all of Denmark is subdivided into so-called ID15 catchments. Each ID15 catchment
represents a topographic basin with an average area of about 15 km$^2$ (outlined in Fig. 1c). The total number of ID15 catchments
is 3351. Out of these, 521 catchments lack a representation of the stream network in the DKM (mostly because they are small
catchments draining directly to the sea) or located in small islands, which have been excluded in this study. With the selected
2830 ID15 catchment, we cover 90.60% of the land area of Denmark. Each of the catchments has data on flow direction and
upstream/downstream catchments, allowing to obtain the total aggregated upstream area for all basins. Fig. 1c shows different
scales of ID15 catchments, each of the shapefiles represents a catchment unit and connects with the upstream routing area. The
catchment boundary to any required points on river networks is defined by identity index of the catchment unit. The ID15
catchments are connected with DKM discharge points (Q points), which are the grid points of the MIKE Hydro River setup
where simulated discharge time series are available (DHI, 2020). Details of Q points will be described in the next section.
Based on the ID15 catchment dataset, we prepared a dataset of catchment attributes and hydrometeorological time series for
the 2830 catchments, like the widely used CAMELS series dataset (Addor et al., 2017; Alvarez-Garreton et al., 2018; Chagas
et al., 2020; Coxon et al., 2020; Fowler et al., 2021; Höge et al., 2023). The dataset includes static catchment attributes, dynamic
variables of climate forcings, streamflow observations, and DKM simulations. Climate forcings have been described in the
former section and include precipitation, temperature, and potential evapotranspiration. DKM simulated streamflow for each
ID15 catchment was extracted from the Q points at the catchment outlets. The other simulations are grid-based
spatiotemporally distributed variables originating from DKM at 500 m resolution, including actual evapotranspiration, average
soil water content, and phreatic depth. They were all spatially aggregated into a time series for each ID15 catchment, including
the entire upstream area. Fig. 2b shows all the Q points in the DKM with a total number over 48,000. Figure 2c shows the
distribution of ID15 catchments and gauging stations. An example of spatially aggregated variables for a catchment in
southeast Jutland is shown in Fig. 1d and Fig. 1e.





### 2.2.2 Climate forcings and basin attributes

The climate data used in this study includes precipitation, mean temperature, and potential evapotranspiration, which were

obtained from the Danish Meteorological Institute (Scharling, 1999a, b). The temporal resolution of the climate data is daily,

the spatial resolution of precipitation is 10 km and 20 km for both temperature and potential evapotranspiration. Precipitation

was corrected based on daily wind speed and temperature to correct for precipitation sensor undercatch (Stisen et al., 2011).

The climate forcings are used as inputs for both the DKM and LSTM models.

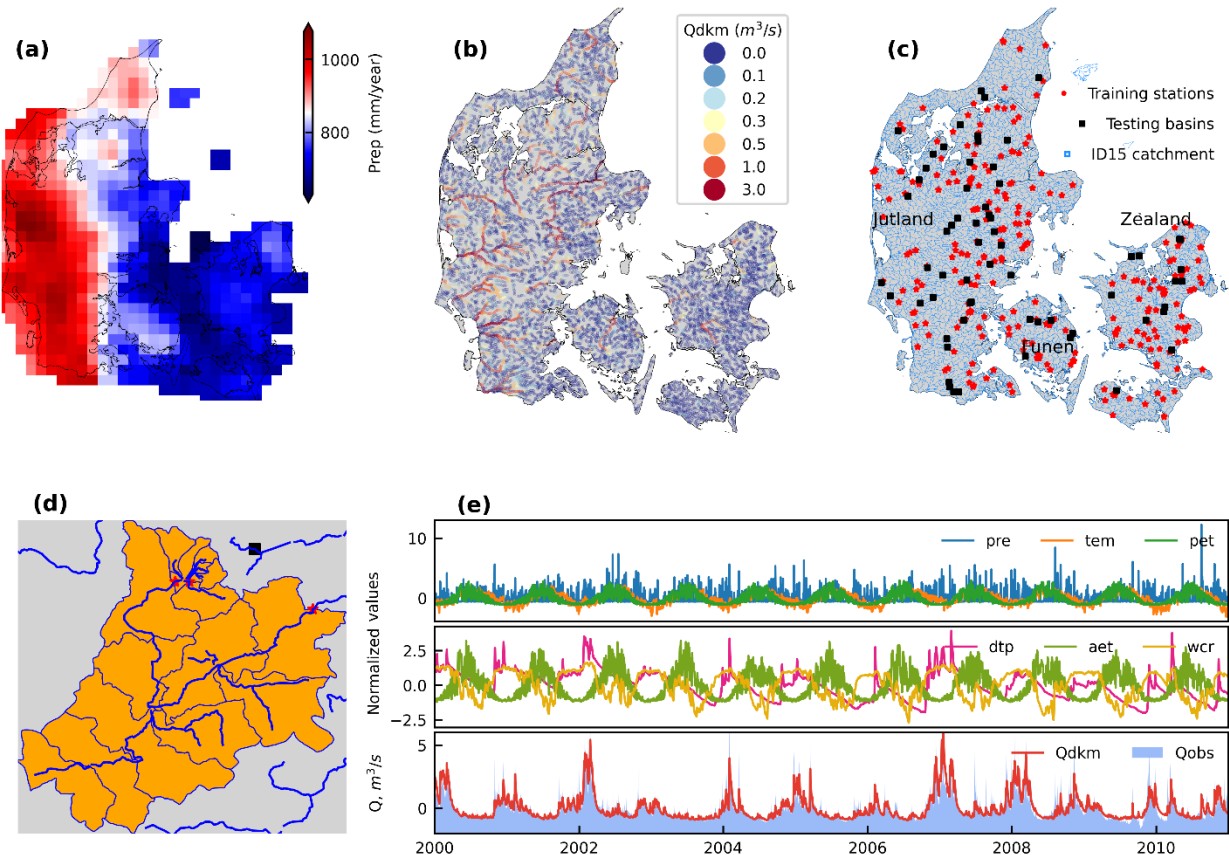


**Figure 1: Hydrometeorological characteristics of the study area. In the subplots, (a) shows the average annual precipitation in Denmark; (b) shows the average streamflow simulated by the DKM; (c) shows the ID15 catchments and the location of gauging stations which have been randomly divided into training (254 stations) and testing stations (64 stations) for LSTM model development; (d) shows the outline of a ID15 catchment (ID: 51350461, Vejle River) and the upstream subbasins which are also**

**included in ID15 catchments; (e) shows spatially aggregated timeseries of the basin (ID: 51350461) during training period, including normalized precipitation (pre), temperature (tem) and potential evapotranspiration (pet), DKM simulated depth to phreatic surface (dtp), actual evapotranspiration (aet), and soil water content (wcr), DKM model simulated streamflow (Qdkm), and observed streamflow (Qobs).**





Catchment attributes, such as land use, soil type, topography, geology, and climate play a pivotal role in hydrological modelling, as variations contribute significantly to the hydrological processes taking place in the basin. We selected 30 static catchments attributes (Table 1) which we consider impacting the hydrological processes in Denmark. The spatial distribution of these attributes is shown in Appendix A. The average elevation of all the catchments ranges from 0.10 m to 144.10 m with a median elevation of 29.70 m. The median slope is 15.10% of all the catchments. The average clay content is higher in the

east than west Jutland. The static catchment attributes include simulation outputs from the DKM: the phreatic depth is included, the median value is -1.91 m from a higher resolution DKM (100 m) and –1.55 m from a coarse resolution model (500 m) (Schneider et al., 2022b; Koch et al., 2021). The spatial distribution of phreatic depth shows it is low in north and middle Jutland. The median value of phreatic depth is -2.23 m in summer, and high in winter with a median value of –1.54 m. Agriculture is the main land use type occupying 29.19%. Southern and central Jutland has a higher chalk aquifer depth, and

clay thickness above chalk aquifer.

**Table 1. Static catchment attributes**

| Short name | Long name | Short name | Long name |
|---|---|---|---|
| Area | Catchment area | Urban | Fraction of urban |
| DEM | Digital elevation model (DEM) | Mean pre | Average precipitation |
| Slope | Slope calculated from DEM | Mean tem | Average temperature |
| Clay_tot_a | Average clay content across A horizon [%] | Mean pet | Average potential evapotranspiration |
| Clay_tot_b | Average clay content across B horizon [%] | Aridity | Ratio of mean PET to mean precipitation |
| Clay_tot_c | Average clay content across C horizon [%] | DKM_q | DKM simulated discharge |
| Clay_tot_d | Average clay content across D horizon [%] | DKM_aet | Actual evapotranspiration (500m model) |
| Dtp | Phreatic depth (100m model) | DKM_wcr | Average soil water content (500m model) |
| Dtp_s | Phreatic depth in summer (100m model) | DKM_dtp | Phreatic depth (500m model) |
| Dtp_w | Phreatic depth in Winter (100m model) | Clay depth | Depth of clay |
| Dtp_1m | 1m exceedance probability of phreatic surface (100m model) | Depth CA | Depth to chalk aquifer |
| Dtp_2m | 2m exceedance probability of phreatic surface (100m model) | Thick CA | Thickness of chalk aquifer |
| Agriculture | Fraction of agriculture | Clay thick CA | Clay thickness directly above chalk aquifer |
| Forest | Fraction of forest | Clay thick ACA | Accumulated clay thickness above chalk aquifer |
| Lake | Fraction of lakes | Chalk transm | Chalk transmissivity |

### 2.2.3 Benchmark model 1 - the National Water Resources Model (DKM)

The DKM has been developed at the Geological Survey of Denmark and Greenland (GEUS) over the course of several decades (Henriksen et al., 2021, 2003; Højberg et al., 2013; Soltani et al., 2021; Stisen et al., 2020). It is built on the MIKE SHE hydrological modelling framework using a transient, fully distributed, physics-based description of the terrestrial hydrological cycle (Højberg et al., 2013; Stisen et al., 2020; Abbott et al., 1986; DHI, 2020), 3D subsurface flow is coupled to processes in





the unsaturated zone, 2D overland flow and surface water routing in streams. The model is run with daily climate forcings (section 2.2.2) and is calibrated against daily streamflow observations from ~300 stations across Denmark (stations shown in Figure 1c), as well as groundwater head observations. It currently exists at two horizontal resolutions, 100m and 500m. For our case, we use the 500m version due to its reduced computational demand and the limited effect of enhanced grid resolution on streamflow simulations. For simulation of streamflow, MIKE SHE is coupled to the surface water model code MIKE Hydro River. In the case of the DKM, simple streamflow routing is applied as focus is on streamflow simulation (DHI, 2020). The

MIKE SHE and MIKE Hydro River models are coupled through river links, where water is exchanged between river channel, land surface and subsurface. In the 500m version of the DKM, approximately 20,000 km of water courses are represented in this manner.

### 2.2.4 Benchmark model 2 - LSTM rainfall-runoff model (LSTM-rr)

LSTM-rr uses meteorological forcings, including precipitation, temperature, and potential evapotranspiration as
dynamic inputs, together with catchment attributes as embedded static inputs when the training and testing basins are more than one, and discharge observed at basin outlets as the target variable to develop the LSTM networks (Fuente et al., 2023; Hashemi et al., 2022; Koch and Schneider, 2022; Kratzert et al., 2021, 2018). The networks are usually trained and tested using historical data from a group of gauged basins and applied to extrapolate streamflow for unmonitored period or ungauged basins. LSTM-rr has gained popularity due to their ability to
capture complex temporal dependencies and nonlinear relationships, and the predicted streamflow has often been found to outperform traditional hydrological models (Hauswirth et al., 2021; Frame et al., 2021a; Lees et al., 2021; Wilbrand et al., 2023; Feng et al., 2020).

### 2.3 LSTM hybrid schemes

We created four LSTM models distinguished by input sequences and target variables as the candidate hybrid model for
streamflow simulations at national scale. The tested models include 1) pretraining-finetuning rainfall-runoff model, 2) dynamic inputs model with DKM simulations and climate forcing, 3) residual error prediction model, and 4) error factor prediction model. The first serves as benchmark to assess the accuracy that can obtained by a standalone LSTM model without a hybrid scheme. The remaining four models represent different implementations of hybrid models. The following subsections describe the details of these models.

### 2.3.1 Pretraining and finetuning LSTM rainfall-runoff model (LSTM-pf)

Pretraining and finetuning are techniques used to improve the performance of neural networks on specific tasks (MacNeil and Eliasmith, 2011; Käding et al., 2017; Cai and Peng, 2021). These techniques are commonly employed in transfer learning, where knowledge learned from one task or dataset is transferred to another related task or dataset (Li and Zhang, 2021; Tan et



al., 2018). Pretraining involves training a neural network on a large dataset or a related task before finetuning it for the target
task. This helps the model learn useful features and representations from the large dataset and grasp general patterns of the
data. Finetuning takes a pretrained neural network and further trains it on a smaller dataset specific to the target task, updating
its weights accordingly. In this study, we pretrained an LSTM-rr model based on all ID15 catchments, climate forcings as
dynamic inputs, basin attributes as static inputs, and DKM simulated streamflow as the target variable. This process enables
the LSTM model to learn major features between climate data and the simulated discharge. Finetuning is then conducted on
basins of observed discharge, i.e., the target variable is changed from DKM simulation discharge to observations. The
hyperparameters are the same for both pretraining and finetuning. The total number of epochs is equivalent to that of LSTM-
rr, with the first half is allocated for pretraining and the second half dedicated to fine-tuning.

### 2.3.2 Hybrid dynamic inputs LSTM model (LSTM-q)

In this configuration, the dynamic inputs are expanded with DKM simulations that impact river streamflow, including depth
of the phreatic surface, average soil water content, actual evapotranspiration, and the DKM simulated streamflow itself. The
depth to phreatic surface varies among basins with different hydrogeological properties, like permeability of the subsurface
materials, aquifers, and confining layers. Groundwater pumping for irrigation, industrial use, or drinking water supply can
significantly alter the interaction between phreatic surface depth and river discharge. Pumping can lead to a lowering of the
groundwater table, reducing the groundwater contribution to river flow. DKM includes water extraction for drinking water
supply and irrigation, thus, the variation of phreatic depth reflects the impacts of climate conditions and human activities.

### 2.3.3 LSTM residual error model (LSTM-qr)

Often, streamflow of a river exhibits strong seasonality due to changes in precipitation and temperature throughout the year.
Simulated streamflow and their associated errors often exhibit systematic patterns such as overestimating baseflow or
underestimating high flow during specific periods and rates. This occurs because of the limitations in model structures and
parameters. The misfitting follows certain regular patterns that can potentially be identified through data-driven algorithms.
Some studies attempted to predict the residuals between PBM simulated streamflow and observations (Cho and Kim, 2022;
Konapala et al., 2020). They argue that the variabilities of residuals are lower in comparison to the variabilities of streamflow
itself, and their results showed that the streamflow simulations could be improved after applying the predicted residuals to
PBMs simulated streamflow.
However, special attention should be paid to the residual time series because data-driven methods cannot effectively learn or
predict them when residuals consistently manifest as random noise. To test the whiteness of residuals between DKM
simulations and observations, we therefore analyse the autocorrelation to ensure that the time series of residuals are not simply
related to noise. Figure 2 illustrates an example of the residuals between simulated and observed streamflow on a daily scale
at the station shown in the previous figure. The residuals were calculated by observed streamflow minus DKM simulations, so





a positive residual indicates that the DKM simulations are lower than observations. It can be observed in Figure 2a that the simulated streamflow is typically underestimated in winter (high-flow seasons)

and overestimated in the warm seasons (low-flow seasons), consistently occurring every year. The autocorrelation figure reveals several spikes outside the 99% bounds, indicating that the time series of residuals are not white noise and could potentially be predicted by LSTM networks.


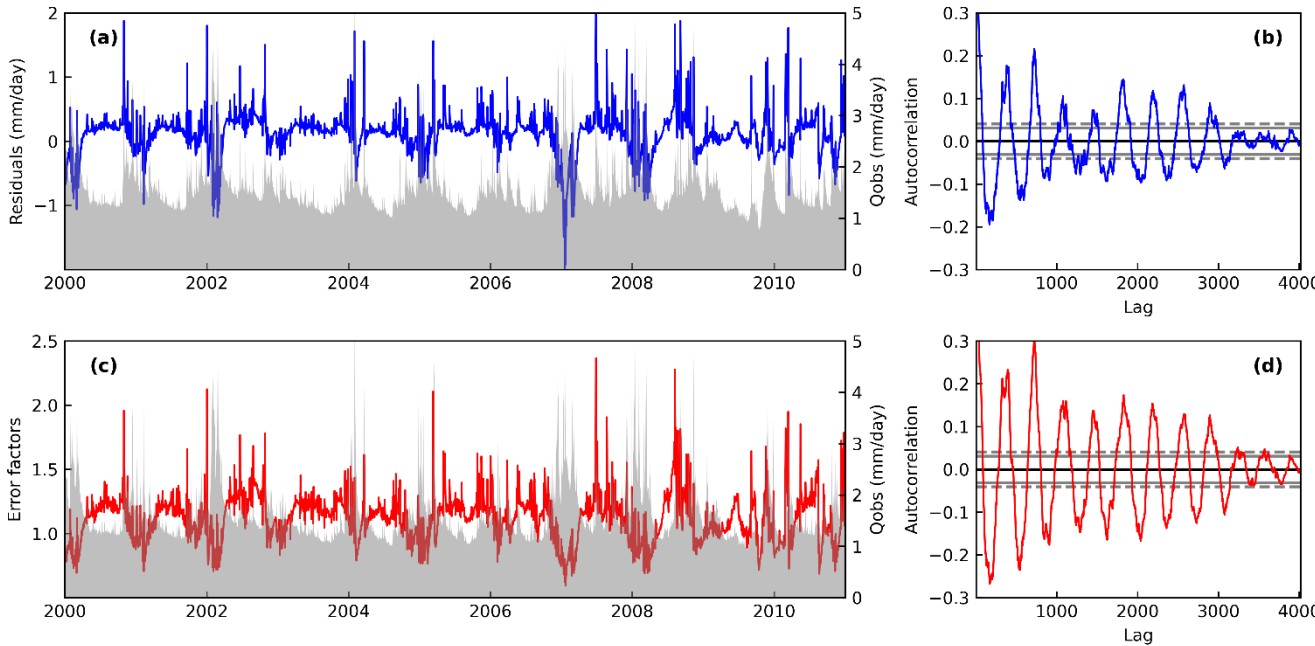

**Figure 2: Time series of streamflow residuals and error factors between DKM simulated streamflow and observations at a hydrological station (ID: 51350461). Autocorrelation of the time series are displayed in the second column to test white noise. The horizontal grey lines in the autocorrelation plots correspond to 95% (dash) and 99% (solid) confidence bands.**


### 2.3.4 LSTM error factor model (LSTM-qf)

The configurations of LSTM-qf are similar to LSTM-qr, but the target variables are relative error factors between observed streamflow and DKM simulations, instead of absolute residuals. The error factors were calculated by dividing observations with DKM simulations, so a value of 1 means DKM simulations are equal to observations. For example (Figure 2), we can see

that DKM underestimates streamflow in winter when the precipitation is high and underestimates the streamflow in summer. Compared to streamflow residuals, error factors exhibit more variability and outliers (Figure 2c). The simulations are over 2 times lower than the observations during high flow events, which could be due to a mismatch in the peak-flow dates. For instance, the error factors are extremely high on one date and drop to values less than 1 on the following day, indicating a





mismatch in the peak-flow times. The plot shows that the error factors in time series are correlated and can be predicted by
data-driven algorithms.

## 2.4 Experiment settings

To assess the potentials of various LSTM hybrid modelling schemes within both gauged and ungauged basins, we conducted
a series of validation experiments. There are 318 gauged basins (Figure 1), which were randomly partitioned into training
basins consisting of 254 stations (80%) and test basins comprising 64 stations (20%). Streamflow was divided into a training
period from 2000 to 2010, the same as DKM calibration period, a testing period from 1990 to 1999, and a validation period
from 2011 to 2019. We followed the design by Koch and Schneider (2022) and created temporal split experiments and
spatiotemporal split experiments to evaluate the performance of LSTM models in gauged and ungauged basins. The temporal
split experiment used the 254 training stations for training during the period from 2000 to 2010, and the same stations were
used for testing during the test period from 1990 to 1999. The spatiotemporal split-sample experiment uses 254 stations for
training during 2000 to 2010, and the trained model was tested on the 64 testing stations during 1990 to 1999.

The neuralhydrology python package is used to train and test all LSTM networks. The package is developed by Kratzert et al.
(2022) and has been widely used in research after it was open-resourced (Frame et al., 2021b; Klotz et al., 2022; Koch and
Schneider, 2022; Nearing et al., 2022; Wilbrand et al., 2023). All the LSTM hybrid schemes are trained with neuralhydrology
package based on PyTorch on a server equipped with a NVIDIA A40 GPU (Paszke et al., 2019). The standard PyTorch
implementation cudaLSTM in neualhydrology package is used for LSTM training due to its efficiency. Dynamic inputs and
static attributes are passed through embedding networks. The optimizer is Adam, and the loss function is Nash-Sutcliffe
efficiency (NSE) for models with streamflow as target variable and root mean square errors (RMSE) for models with residuals
or error factors as target variables.

**Table 2. The potential values of hyperparameters for LSTM models**

| Hyperparameter | Number of epochs | Hidden unit size | Dropout rate | Batch size | Learning rate | Length of sequence |
|---|---|---|---|---|---|---|
| Potential values | [15, 20, 25, 30] | [64, 128, 256] | [0.1, 0.3, 0.5] | [128, 256, 512] | [$10^{-3}$, $5*10^{-4}$, $10^{-4}$] | [60, 90, 180, 365, 730] |

Before using LSTM networks for specific tasks, it is necessary to determine the values of critical hyperparameters. Since there
is no standard method to find an optimal set of hyperparameters for our case, we selected relevant hyperparameters based on
previous studies and assessed their sensitivity (Cho and Kim, 2022; Hashemi et al., 2022; Kratzert et al., 2018). The selected
hyperparameters include the number of training epochs, the size of hidden units, dropout rates, batch size, learning rates, and
the lookback length of the sequence. The tested values for these hyperparameters are defined in Table 1. To assess the
performance of all candidate hyperparameter combinations, a total of 1620 (4*3*3*3*5) possible combinations were
generated. It is challenging to test all combinations for different LSTM models due to limited computational resources. Hence,





we randomly selected hyperparameters from the ranges listed in Table 2 and created 100 candidate hyperparameter combinations. The combination demonstrating the highest performance in terms of the average NSE values in the spatiotemporal split-sample experiment will be chosen to configure the final LSTM models. Table 3 shows the final hyperparameters for the LSTM models.

**Table 3. Optimal hyperparameters for LSTM models**

| LSTM models | LL | HS | BC | LR | NE | DR |
|---|---|---|---|---|---|---|
| LSTM-rr | 365 | 64 | 128 | 0.001 | 15 | 0.1 |
| LSTM-q | 90 | 256 | 128 | 0.001 | 25 | 0.5 |
| LSTM-qr | 365 | 64 | 128 | 0.001 | 15 | 0.3 |
| LSTM-qf | 365 | 256 | 128 | 0.001 | 15 | 0.5 |

*NE: number of epochs, HS: size of hidden units, DR: dropout rate, BS: batch size, LR: learning rate, LL: length of lookback sequency.

**2.5 Model evaluations**

A set of statistical metrics is used to assess the performance of DKM and LSTM models (Table 3). These metrics have been
widely used to compare the differences between simulated hydrography and observations (Baroni et al., 2019; Kratzert et al., 2018, 2021; Liu et al., 2022). NSE compares simulations to the average observations, quantifying the proportion of observed variance that the model can explain (Gupta and Kling, 2011). NSE ranges from negative infinity to 1, with 1 indicating a perfect match between model predictions and observations. Logarithmic NSE (NSElog) and squared NSE ($NSE^2$) are two transformations of NSE; the former applies a logarithmic transformation to discharge before calculating NSE, while the latter
applies a square transformation. NSE emphasizes errors associated with peak flows, $NSE^2$ amplifies extreme values and emphasizes the performance during peak flows even more (Schneider et al., 2022a). NSElog places higher emphasis on errors associated with low-flow situations (Roy et al., 2023). Kling-Gupta efficiency (KGE) combines three components: correlation, bias ratio, and variability ratio, which provides a balanced and comprehensive assessment of model performance. Root Mean Square Error (RMSE) measures the average magnitude of the differences between simulations and observations, which is
suitable for the assessment extreme events modelling.

Further diagnostic signature measures are included to evaluate the performance of simulated streamflow. The high-segment volume (FHV) reflects the 2% peak flow bias of the flow duration curve, the low-segment volume (FLV) reflects 30% low flow bias, and the midsegment slope (FMS) reflects the percent bias of the midsegment slope of the flow duration curve. Peak-timing reflects the time difference between the simulated peak flow and the observed peak flow. Details about these signature
measures are explained in Yilmaz et al. (2008), Gupta et al. (2009), and Kratzert et al. (2021). Table 3 displays the definitions and equations of the above-mentioned indices.





**Table 4. Overview of hydrography evaluation metrics**

| Short name | Long name/ description | Range of values | Best fit value | Reference |
|---|---|---|---|---|
| NSE | Nash-Sutcliffe efficiency | $(-\infty, 1]$ | 1 | Nash and Sutcliffe (1970) |
| KGE | Kling-Gupta efficiency | $(-\infty, 1]$ | 1 | Gupta et al., (2009) |
| RMSE | Root Mean Square Error | $[0, +\infty)$ | 0 | - |
| $NSE_{log}$ | Logarithmic NSE | $(-\infty, 1]$ | 1 | Gupta et al., (2009) |
| $NSE^2$ | Square root NSE | $(-\infty, 1]$ | 1 | Gupta et al., (2009) |
| FHV | High flow volume bias (2%) | $(-\infty,+\infty)$ | 0 | Yilmaz et al., (2008) |
| FLV | Low flow volume bias (bottom 30%) | $(-\infty,+\infty)$ | 0 | Yilmaz et al., (2008) |
| FMS | Middle flow slope bias (20% and 80%) | $(-\infty,+\infty)$ | 0 | Yilmaz et al., (2008) |
| Peak timing | Mean peak time lag (in days) | $[0, +\infty)$ | 0 | Kratzert et al., (2021) |

**Table 5. Performance of DKM and the LSTM hybrid models in temporal split experiment and spatiotemporal split experiment (in bold).**

| | | Temporal split experiment | | | | | | Spatiotemporal split experiment | | | | | |
|---|---|---|---|---|---|---|---|---|---|---|---|---|---|
| | | DKM | LSTM-rr | LSTM-pf | LSTM-q | LSTM-qr | LSTM-qf | DKM | LSTM-rr | LSTM-pf | LSTM-q | LSTM-qr | LSTM-qf |
| NSE | Mean | 0.58 | 0.80 | 0.72 | 0.80 | 0.81 | 0.77 | 0.52 | 0.59 | 0.52 | 0.63 | 0.63 | 0.62 |
| | **Median** | **0.65** | **0.84** | **0.78** | **0.84** | **0.85** | **0.81** | **0.59** | **0.68** | **0.65** | **0.70** | **0.73** | **0.71** |
| KGE | Mean | 0.65 | 0.78 | 0.70 | 0.81 | 0.83 | 0.80 | 0.59 | 0.61 | 0.54 | 0.65 | 0.64 | 0.65 |
| | **Median** | **0.70** | **0.81** | **0.72** | **0.83** | **0.86** | **0.83** | **0.62** | **0.64** | **0.57** | **0.70** | **0.68** | **0.69** |
| $NSE_{log}$ | Mean | 0.53 | 0.76 | 0.61 | 0.74 | 0.76 | 0.76 | 0.41 | 0.36 | 0.37 | 0.47 | 0.48 | 0.49 |
| | **Median** | **0.66** | **0.82** | **0.69** | **0.80** | **0.83** | **0.81** | **0.58** | **0.63** | **0.49** | **0.61** | **0.66** | **0.63** |
| $NSE^2$ | Mean | 0.12 | 0.65 | 0.57 | 0.65 | 0.64 | 0.52 | 0.15 | 0.46 | 0.38 | 0.46 | 0.43 | 0.44 |
| | **Median** | **0.39** | **0.70** | **0.61** | **0.69** | **0.71** | **0.62** | **0.41** | **0.53** | **0.49** | **0.57** | **0.56** | **0.52** |
| FHV | Mean | 0.44 | -3.59 | 1.36 | 1.39 | 2.42 | 3.28 | -0.74 | -5.66 | 4.64 | 0.28 | 1.52 | 1.14 |
| | **Median** | **0.36** | **-3.21** | **0.65** | **1.54** | **2.34** | **3.31** | **-1.32** | **-3.64** | **7.77** | **0.61** | **1.20** | **2.30** |
| FLV | Mean | 108.23 | 47.62 | 144.82 | 91.56 | 47.57 | 57.11 | 84.23 | 63.96 | 139.02 | 127.60 | 52.97 | 73.75 |
| | **Median** | **35.42** | **14.95** | **61.92** | **29.42** | **16.61** | **24.88** | **13.52** | **17.62** | **42.25** | **31.46** | **1.10** | **10.71** |
| FMS | Mean | -1.39 | -10.65 | -18.55 | -13.20 | -7.49 | -9.87 | 7.13 | -8.44 | -16.50 | -10.01 | 2.73 | -4.24 |
| | **Median** | **-6.53** | **-10.31** | **-20.21** | **-12.61** | **-8.46** | **-10.45** | **2.79** | **-18.17** | **-22.73** | **-19.09** | **-7.78** | **-8.35** |
| Peak timing | Mean | 0.91 | 0.60 | 0.64 | 0.65 | 0.67 | 0.79 | 0.82 | 0.56 | 0.53 | 0.67 | 0.64 | 0.71 |
| | **Median** | **0.80** | **0.56** | **0.56** | **0.58** | **0.60** | **0.69** | **0.79** | **0.43** | **0.44** | **0.53** | **0.57** | **0.64** |

## 3 Results

### 3.1 Long-term performance of LSTM hybrid schemes

The cumulative distribution function (CDF) of evaluation metrics for the temporal split experiment (subplots with white backgrounds) and spatiotemporal split experiment (subplots with gray backgrounds) are shown in Fig. 3. Mean and median values of the evaluation metrics are listed in Table 5. In general, all LSTM models outperformed the DKM, underlining the potential of utilizing LSTM models for streamflow estimation. LSTM-qr (mean NSE is 0.81) exhibits the best model performance, closely followed by LSTM-q (median NSE is 0.80), LSTM-rr (0.80), LSTM-qf (0.77), and LSTM-pf (mean NSE

is 0.73) in the temporal split experiment. LSTM hybrid models unaltered the performance significantly compared with the benchmark model LSTM-rr.

Performance of all LSTM models decreased when applied to ungauged basins, as revealed by the spatiotemporal split experiment. LSTM-q slightly outperforms LSTM-qr according to NSE and $NSE^2$ in the spatiotemporal split experiments, indicating that LSTM-q is more effective for high-flow modelling. This is further supported by FHV, which measures the bias





of peak flow where LSTM-q shows a lower error compared to LSTM-residual. In contrast, LSTM-qr demonstrates higher

performance at low flows conditions with higher NSElog and lower FLV bias. The DKM model exhibits a higher peak timing

error, while LSTM-rr shows the lowest peak timing error. LSTM-rr shows a lower NSElog than DKM, LSTM-q, and LSTM-

qr, indicating its accuracy over low flow is poorer.




**Figure 3: Performance of benchmark models and LSTM hybrid models in temporal split experiment (subplots with white background) and spatiotemporal split experiment (subplots with grey background).**

Figure 4 shows the spatial distribution of NSE of DKM at all stations and the enhancements in NSE achieved by LSTM hybrid

modelling. DKM exhibits satisfactory performance in most basins, only seven stations from the temporal split experiment (out

of 254 total) and five stations (out of 64 total) from the spatiotemporal split experiment display a negative NSE. DKM has

difficulties in modelling streamflow in basins covered by large lake areas (Fig. 4), such as stations situated in central Jutland

and northeast Zealand. LSTM hybrid models have improved NSE at many stations, as illustrated in the histogram in Fig. 4b-





f. Specifically, LSTM-rr has shown improved NSE at 40 stations in the spatiotemporal split experiment, LSTM-q improved

43 stations, LSTM-qr improved 47 stations, and LSTM-factor improved 48 stations. However, LSTM hybrid schemes still fail

to enhance NSE at some stations, such as 24 stations for LSTM-rr, 21 stations for LSTM-q, and 17 stations for LSTM-residuals.

**Figure 4: Performance of DKM and LSTM models during the testing period (1990-1999) of temporal split experiment (marked by**
**star) and spatiotemporal split experiments (marked by square). (a) NSE of DKM. The histogram can be understood as legend to the**
**map and the width bars indicate the number of testing stations in corresponding ranges of NSE. (b – f) shows the differences of NSE**
**between DKM and LSTM ($\Delta NSE = NSE_{LSTM} - NSE_{DKM}$). The histogram can be understood as legend to the map and the bars**
**indicate the number of testing stations in corresponding ranges of ΔNSE.**

Figure 5 presents the time series of streamflow for two example stations from the spatiotemporal split experiment located in

the central Jutland, which we have named basin A (ID=13261645) and basin B (ID=35324466). The DKM model





overestimates high-flow periods and underestimates low-flow periods in basin A, resulting in a negative NSE. LSTM-rr agrees well with observations during high-flow seasons but tends to overestimate during low-flow periods. The simulated hydrograph of the LSTM hybrid models, falls between the ranges of DKM and LSTM-rr, indicating superior performance compared to

DKM and LSTM-rr. However, the finding differs in basin B, where DKM-simulated streamflow aligns well with observations, and NSE exceeds 0.6. Basin B is spatially close to basin A, and the climate forcings are equivalent. We then compared the basin attributes of basins A and B with those of the basins used for LSTM training. The slope of basin B (5.03) is significantly higher than that of basin A (1.14) and most training basins (ranging from 0.258 to 4.580). The forest ratio of basin B is 27.61%, whereas it is 5.98% for basin A. These distinct differences between basin A and the training dataset result in the inferior

performance of LSTM models. These results demonstrate the challenges of extrapolating streamflow to ungauged basins and the importance of selecting training datasets with diverse catchment attributes.

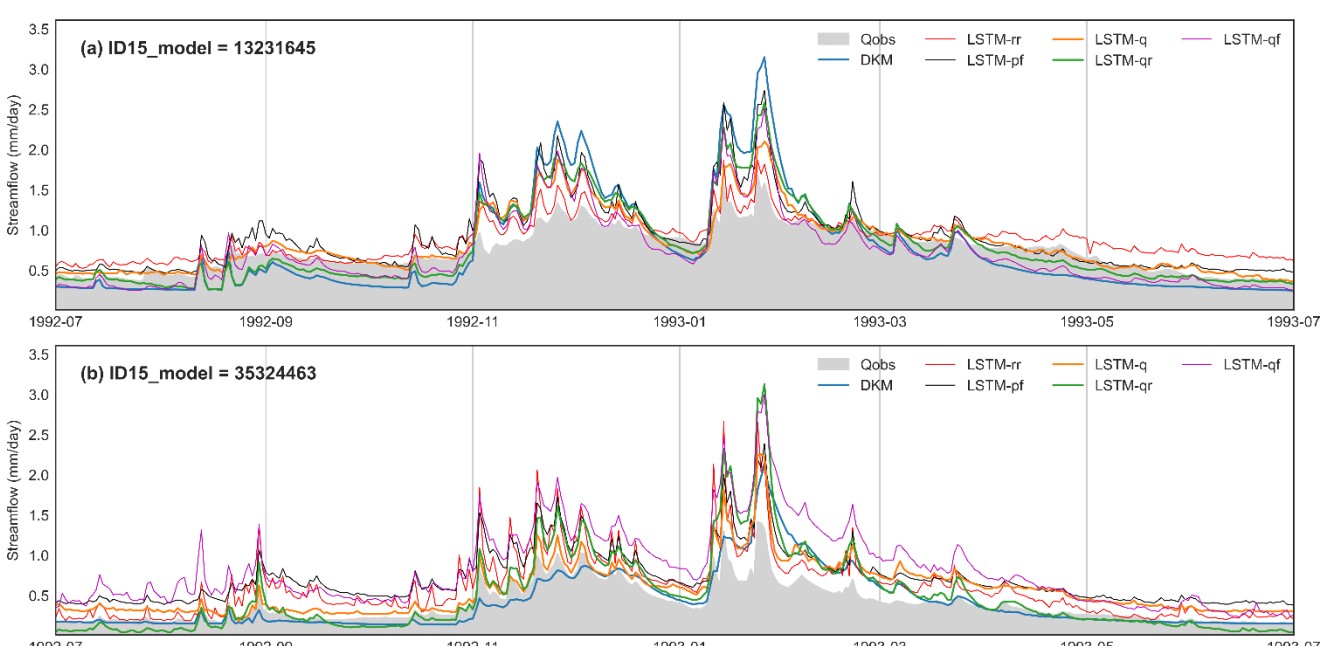

**Figure 5: Time series of streamflow at two hydrological stations which was involved in the spatiotemporal split experiment.**


Fig. 6 presents a heatmap of correlation coefficients between model performance (NSE and ΔNSE) of the different models and static basin attributes. Unsurprisingly, basin area positively correlates with all models' performance, i.e., performance generally is better for larger basins. DKM simulated groundwater levels (dtp, dtp_s, dtp_w, dtp_1m, and dtp_2m) positively correlate with NSE for all models, indicating that the models generally struggle to accurately simulate streamflow in basins

with deeper groundwater levels. In Denmark, much of the streamflow is generated as baseflow; thus, controlled by groundwater levels. With deeper groundwater levels, accurate representation of groundwater level dynamics becomes more challenging.





The negative correlation between model performance and the share of lake area can be explained by the complex interactions in lake water balances; something both the DKM and the LSTM models struggle with. Similarly, increased urban share decreases model performance; again, likely due to complexities and heterogeneities in urban hydrology inadequately

represented in the models. Geological features such as depth to the chalk aquifer, clay thickness above the chalk aquifer, and accumulated clay thickness above the chalk aquifer negatively correlate with the performance of both DKM and LSTM models. The reasons for this require further investigation.

The changes in performance of the LSTM models compared to the DKM ($\Delta$NSE) exhibit a negative correlation with basin area, suggesting that LSTM model improvements decrease with increasing basin size (Fig. 6b). This might be related to all

basin information being aggregated across each basin for the LSTM models, whereas the distributed nature of the DKM allows representation of more complex streamflow generation processes (and routing) within basins. $\Delta$NSE indicates a positive correlation with DKM_wcr and DKM_aet, both showing similar spatial patterns (refer to Appendix A), signifying improved performance of LSTM models over DKM in, generally speaking, basins with higher soil moisture. In such basins, runoff generation might be more driven by complex hydrological and land-surface processes e.g. occurring in wetlands, instead of

more simply being driven by precipitation. The description of such land-surface processes in the DKM, where the simple "2-Layer-method" is being used, are inadequate for capturing some complexities (DHI, 2020; Yan and Smith, 1994). Similarly, the LSTM models show performance improvements for catchments with higher share of lake areas. Again, the representation of lake water balances and streamflow through lakes is one of the weaknesses of the DKM.

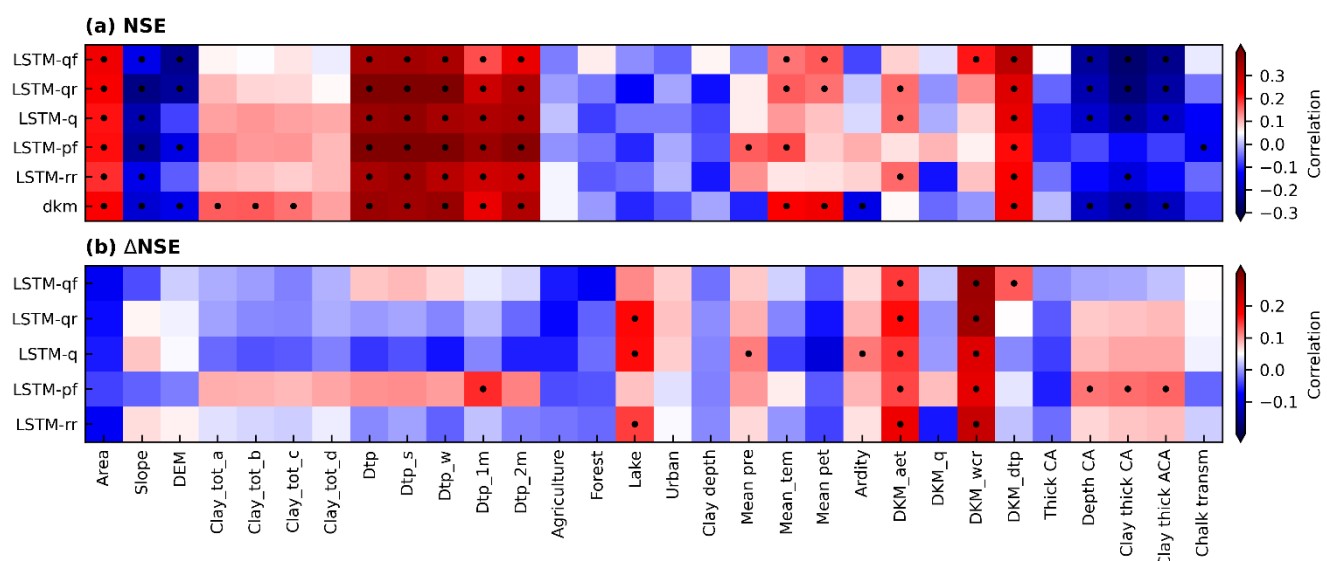

**Figure 6: Correlations between the performance (NSE) and changes in performance ($\Delta$NSE = NSE_lstm - NSE_dkm) of different LSTM models and catchment static attributes. The black points indicate the correlations pass the 95% significant tests.**





### 3.2 Events performance of LSTM hybrid schemes

The objective of developing different LSTM models is to identify an optimal hybrid scheme to support the operational
modelling and forecasting framework, which the DKM is already a part of. A real-time module has been established to collect
daily observations of climate forcings, including precipitation, temperature, and potential evapotranspiration, which serve as
inputs for a real-time DKM. Within the operational real-time framework, emphasis is placed on modelling extreme events.
Therefore, in this section, we investigate the performance of LSTM hybrid schemes in modelling extremely high and low
flows.

We selected four distinct wet periods characterized by high peak flows across many regions of Denmark, as well as two dry
periods marked by severe drought conditions. Figure 7 displays the observed streamflow and simulations from the DKM,
LSTM-q, and LSTM-qr averaged across all stations, as well as the histogram of RMSE for all stations. The two LSTM hybrid
models (chosen based on their superior performance) show improved RMSE compared to the DKM at most stations but fail
at a few stations as indicated by the tail of the fitted frequency density curve (Fig. 7a). Capturing peak flows accurately proves
challenging for both DKM and the LSTM hybrid schemes, as the simulated streamflow values tend to be lower than
observations during the four flooding events.

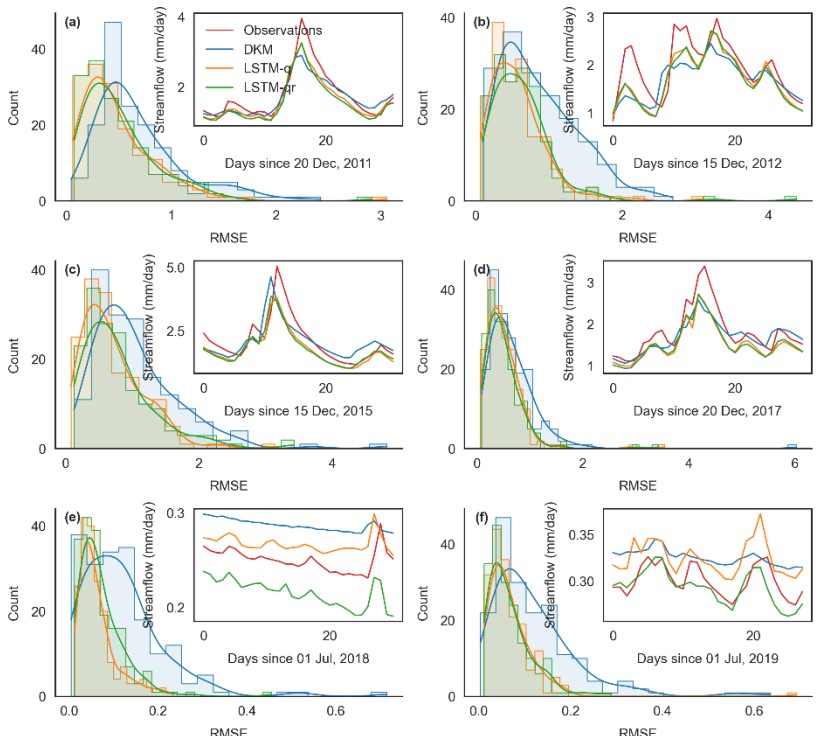

**Figure 7: Performance of DKM, LSTM-q, LSTM-qr during extreme events. (a - d) four flooding events, and (e -f) two drought
events. In each subplot, the main figure shows the histogram of RMSE calculated across all stations and the fitted probability density
function, an additional figure in the top-right shows the averaged time series of streamflow.**



### 3.3 Comparison of LSTM models and DKM at a national level

After developing and identifying the optimal LSTM hybrid schemes, we extended their application from predicting streamflow in gauged basins to ungauged basins, such as the outlets of all ID15 catchments across Denmark. Fig. 8 illustrates the
high/median/low flow of DKM in all ID15 catchments from 2010 to 2020, the residuals between LSTM-q and DKM, and the residuals predicted by LSTM-qr in all ID15 catchments. DKM and LSTM models generally agree well with each other in most basins, with percentage differences close to 0. This further underlines the robustness of the LSTM models, also in spatial extrapolation, as they manage to follow the simulated streamflow patterns from the DKM which is based on a spatially consistent setup, calibrated jointly for all of Denmark. However, discrepancies arise in certain basins, as indicated by deep red
and deep blue colours, particularly during high and low flow conditions. In Jutland, the LSTM models tend to simulate higher low flows compared to DKM, while in Zealand and Funen, the opposite pattern is observable. In western Jutland, where precipitation is higher and DKM-simulated streamflow is larger than in other regions, the LSTM models predict lower high flows. DKM overestimate high flows, and reducing the value of DKM simulations can enhance accuracy in these cases.





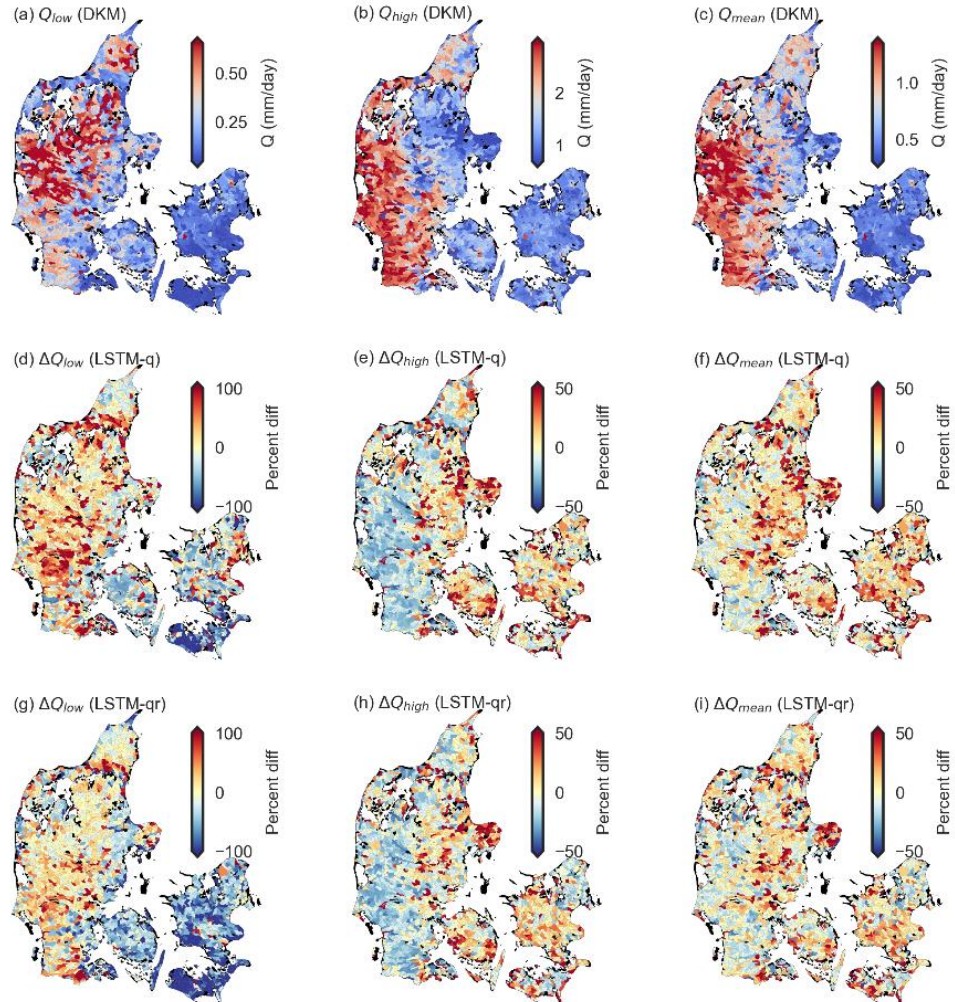


**Figure 8: A comparison of simulated streamflow differences between DKM and two LSTM models (LSTM-q and LSTM-residual). The first row depicts DKM simulated streamflow during high flow, low flow, and median flow conditions, the second row is the bias between DKM simulations and the LSTM-q predictions, and the third row is the bias between DKM and LSTM-qr. The percent diff in the figure is defined as the differences between LSTM model and the DKM, percent diff = $(Q_{LSTM} - Q_{DKM}) \times 100$.**


## 4 Discussion

In this study, a series of experiments were conducted to enhance the performance of streamflow estimation at national scale in Denmark. The main objective was to assess various configurations of LSTM models to identify the optimal configuration to serve as a hybrid model for streamflow prediction. The results revealed that utilizing LSTM models, especially the hybrid

schemes that were coupled with physically based simulations, exhibited superior performance for both long-term periods
(spanning a decade) and short-term extreme events (30 days).

Overall, we found that the trained LSTM models were robust, and their performance was relatively consistent across the tested
hyperparameters. Figure 9 underlines that the variations of NSE across the sensitivity analysis of 100 hyperparameter
combinations are small. Previous studies often applied default hyperparameters for LSTM development, a practice that remains
justifiable due to the generally limited impact of hyperparameter adjustments. However, it is necessary to mention that the
robustness of LSTM models can be further enhanced through the incorporation of physical knowledge into the selection of
hyperparameters. For instance, the selection of a lookback length for sequential time series data traditionally adheres to 365
days for LSTM rainfall-runoff models, a choice made to account for the seasonal dynamics in hydrological processes.
Nevertheless, the lookback length can be reduced to under three months in the hybrid modelling schemes, as model
performance remains reliably consistent across these diverse temporal scales. This suggests that in this case, the longer-term
hydrological information is contained in the PBM outputs such as groundwater levels. Conversely, we find that the LSTM-rr
model, without the DKM as input, benefits from a prolonged lookback length.

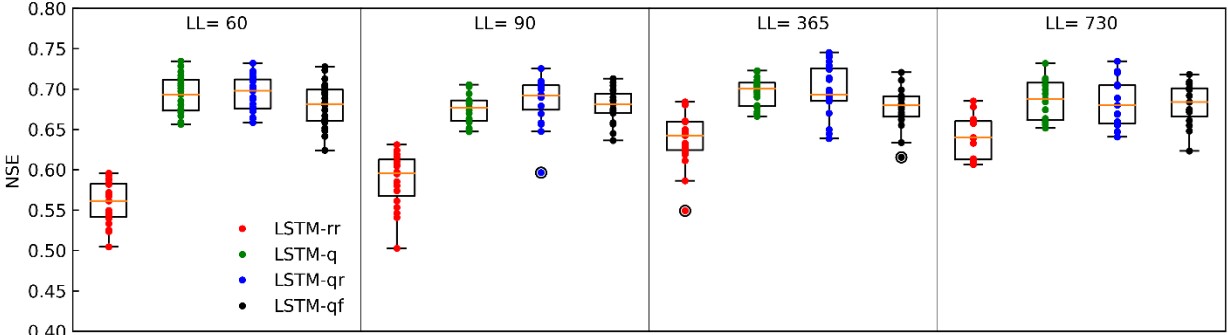

**Figure 9: The relationship between NSE and sequential lookback length (LL) of the spatiotemporal split experiment.**

The design of the applied temporal split experiment and spatiotemporal split experiment aimed at illustrating the potential
performance of LSTM models in gauged and ungauged basins. In this study, the performance of LSTM models (NSE > 0.8)
in gauged basins are comparable to previous studies (Cho and Kim, 2022; Lees et al., 2021; Konapala et al., 2020; Frame et
al., 2021c). The performance dropped for ungauged basins (spatiotemporal split experiment in Fig. 3), well aligned with (Koch
and Schneider, 2022).  Few studies conducted a comparable spatiotemporal hold-out experiment, thus special attention should
be paid to validate the performance of LSTM models over ungauged basins. For example, Kratzert et al. (2019b) applied a 12-
fold cross validation experiment over the contiguous United States and found a limited drop in performance for the predictions
in ungauged basins. However, the applied 8.3% spatial holdout may not pose the most challenging validation test. In our study,





we applied a larger 20% spatial hold-out and a more systematic k-fold validation test was hampered by inconsistent length of observations across the Danish discharge stations.

The intricate interactions between groundwater and surface water have posed challenges for simulating streamflow using rainfall-runoff models in many basins of Denmark (Danapour et al., 2019; Duque et al., 2023). We tested LSTM-rr for streamflow estimation, and the results were encouraging, with the mean Nash-Sutcliffe Efficiency (NSE) improving from 0.58

(DKM) to 0.80 (Table 5). These improvements indicate the large potential of LSTM-rr model for streamflow modelling. However, it is important to note that LSTM-rr may not perform well everywhere, as evidenced by its limitations in strongly groundwater dependent regions, such as northern Jutland. LSTM-rr simulates quick responses to the variations of precipitation well but can fail to predict reduced baseflows due to depleted groundwater storage (Fig. 5a). Also, the performance drop between temporal and spatiotemporal holdout is most pronounced for LSTM-rr (NSE is reduced from 0.80 to 0.59). Therefore,

it is important to emphasize the advantages of integrating physical data into the LSTM framework, and the adoption of hybrid schemes, such as LSTM-q and LSTM-qr, yielded improvements in the estimation of streamflow. These results align with the findings of previous studies (Feng et al., 2022; Frame et al., 2021c; Hunt et al., 2022; Zhang et al., 2023; Cho and Kim, 2022; Konapala et al., 2020; Tang et al., 2023) that assessed the potential of hybrid modelling.

We tested four different hybrid systems: LSTM-pf, LSTM-q, LSTM-qr, and LSTM-qf. They all exhibited satisfactory

performance for streamflow estimation according to the evaluation metrics, with the order of priority (from high to low) being LSTM-qr ≈ LSTM-q > LSTM-rr > LSTM-qf > LSTM-pf ≈ DKM. The better performance of LSTM-q is consistent with previous studies, for instance, Cho and Kim, (2022) proved that WRF-Hydro-LSTM has a lower percent bias than LSTM-rr. Tang et al. (2023), Frame et al. (2021b), and Hunt et al. (2022) showed that LSTM models with additional datasets of hydrological signals as inputs as well as simulations of global hydrological models outperformed LSTM-rr. However, our

finding that LSTM-qr is slightly better than LSTM-q differs from Konapala et al., (2020). They pointed out that the LSTM-qr model was inferior to LSTM-q across the conterminous US. In their work, LSTM-qr showed comparable performance with LSTM-q when the NSE of PBM was larger than 0.75, and the improvement of LSTM-qr then decreased as the NSE of PBM decreased. Thus, the performance of LSTM-qr was overly constrained by the performance of the underlying PBM, whereas the LSTM-q was found to be more flexible. In our study, DKM performs better than the PBM in Konapala et al., (2020), and

27% of the stations have an NSE higher than 0.75, whereas the percentage is 18% in their study. Thus, this can explain the slightly increasing performance of LSTM-qr compared with LSTM-q in our case, because the underlying PBM, the DKM, performs generally very well. Additionally, the performance of LSTM-pf is not comparable to the other LSTM hybrid schemes, which differs from the conclusion of Koch and Schneider (2022). This can be explained by the fact that in the pre-training, the model is pre-trained against DKM simulated streamflow from all 2830 ID15 catchments as the target variable, whereas the

finetuning is performed against only the observation station data. This may introduce more complexity and noise for LSTM to learn. Koch and Schneider (2022) only pre-trained using simulated DKM based streamflow at the same basin where observations were available. We also implemented an experiment that pre-trained a model on gauged basins only with DKM simulated streamflow as target variables, then finetuning the model with observations, and the performance is comparable to




LSTM-rr. To our knowledge, LSTM-qf is a novel hybrid modelling scheme, tested for the first time in the present study. The
performance of LSTM-qf is lower than LSTM-qr. This is likely related to the use of DKM simulated streamflow as
denominator when calculating the error factors, which can be problematic if simulated streamflow is close to zero resulting in
large and instable factors. Figure 2 shows that the variability of error factors is larger with more outliers than residual time
series. Thereby, we recommend for future work to focus on the residual approach instead of the factor approach.

We intended to train a skillful LSTM model to be used to forecast discharge across Denmark in an operational real-time
framework, currently under development. However, the LSTM networks presented in this study have shallow structures and
trained against a limited number of gauged basins, limiting their ability to capture some features deeply hidden in the
hydrometeorological time series and catchments attributes. The catchments have a large variety of static attributes spatially,
and the hydrological regimes change significantly across Denmark. The hybrid schemes alleviated the shallow structures as
discussed above, other solutions could be enhancing the complexity of the neural networks, and developing specific DL models
for different regions distinguished by regime information (Hashemi et al., 2022).

Spatially, we predicted streamflow at a large number of catchments, namely 2830 outlets, covering most of Denmark. The
comparison of LSTM and PBM performance across the entire region gives some insights in controlling factors on the different
models' performance, potentially guiding further model improvement (especially of the PBM). Another question that arises in
this case of nested catchments is how LSTM models can be developed that produce consistent streamflow simulations along
river courses, with as many Q points as distributed hydrological models. This is particularly useful, as many PBMs currently
provide streamflow simulations at explicit grids or points within the catchment (Harrigan et al., 2023). Correcting the
streamflow at each PBM simulation point offers advantages, such as improving the prediction of local flooding extent,
assessing drought hazards, and estimating nitrate transport, all of which require a refined resolution of streamflow at local
scales. This is why LSTM-qr and LSTM-qf hybrid schemes were considered in this study, which can be predicted at the basin
outlets and, potentially, can be applied to all Q-points within a subbasin. Ideally, discharge routing in the river channel involves
linear accumulation from upstream to downstream and therefore, we can use relative residuals or error factors not only at basin
outlets but also for upstream locations. However, implementing such an idea is challenging, given that river routing processes
do not change linearly from upstream to downstream due to additional water from small tributaries, groundwater contributions,
and river regulation. Further information on river routing and the relationship of streamflow between upstream Q points and
outlets should be considered, and advanced methods should be investigated for distributing residuals and error factors to all
the Q points upstream. On the other hand, the development of distributed LSTM rainfall-runoff models or distributed LSTM
hybrid schemes could be a new topic in the future.

## 5 Conclusion

This study aimed at identifying optimal LSTM hybrid schemes based on the National Water Resources Model (DKM) to
enhance streamflow estimation at a national scale. To achieve this, we developed four LSTM hybrid models with varying





dynamic inputs and target variables, evaluating them under different scenarios, including temporal and spatiotemporal split experiments. Two optimal LSTM models, LSTM-q and LSTM-qr, were further assessed for their performance in extreme events. Lastly, we compared the disparities between DKM and the optimal LSTM models, seeking insights into hydrological modelling from both perspectives. The key conclusions of this study are:

(1) LSTM models excel at modelling streamflow in Denmark, demonstrating superior performance compared to DKM. The LSTM-rr model performs satisfactorily in numerous basins, with a mean NSE of 0.80 in the temporal split experiment and 0.59 in the spatiotemporal split experiment. However, it faces challenges in simulating streamflow in groundwater-dominant regions as well as spatial transferability, which can be mitigated by employing hybrid LSTM models.

  (2) The best-performing hybrid models are LSTM-qr and LSTM-q, achieving mean NSE values of 0.80 and 0.81, respectively.

Also, in ungauged basins they surpass the DKM performance, with a mean NSE of 0.63, compared to 0.52 of the DKM. In the spatiotemporal split experiment, LSTM-qr improved the accuracy compared to the DKM for 73% of stations, while LSTM-q improved 67%. Basin attributes such as catchment area, average clay content, and phreatic depth correlate positively with model performance, whereas factors like slope, DEM, lake ratio, depth, and clay thickness related to chalk aquifers correlate negatively with model performance.

(3) LSTM hybrid models also contribute to improving the modelling of extreme events. LSTM-qr and LSTM-q effectively reduce errors in DKM simulated values during high and low-flow periods in Denmark. But still, more efforts should be made to improve the modelling accuracy toward extreme values in the hydrographs, considering LSTM models underestimate the peak flow of flooding events.

  The utilization of LSTM in river streamflow modelling heralds a promising perspective for hydrological predictions. Previous

studies focused more on gauged basins, while this study contributes to the topic with a national scale analysis. We found that the conventional LSTM-rr model has limited performance in regions with complex hydrological processes. Information from physical hydrological models is helpful, as indicated by the benefits across several hybrid schemes. Our future plans include evaluating the hybrid schemes in a real-time forecasting framework forced by forecasted climate data and developing distributed LSTM hybrid schemes.


*Code and data availability.* The LSTM code used in this study is based on the neuralhydrology Python package, which can be accessed via https://neuralhydrology.github.io. Upon request, the corresponding author will provide scripts for data preprocessing, post-processing, and visualization. The climate forcings and DKM simulations, as well as the in-situ observations used in this study, are currently being organized for publication and are available upon request prior of publication

of the data paper.

*Author contributions.* All authors contributed to develop the original idea and design experiments. JL conducted the experiments, in close cooperation with RS and JK, and further inputs from SS and LT. The code for data preprocessing, LSTM



training and testing, data post-processing and visualization were prepared by JL with support by RS and JK. JL and RS prepared
the manuscript, with contributions from JK, SS, and LT. All authors were involved in writing the manuscript.

*Competing interests.* The authors declared that there are no competing interests.

*Acknowledgements.* The work presented in this manuscript was performed to enhance flood warning for Denmark as part of
the establishment of the Danish flood warning system (varslingssystem for oversvømmelser) from 2023 to 2026. The project
is being led by the Danish Meteorological Institute (DMI), in cooperation with the Geological Survey of Denmark and
Greenland (GEUS), the Danish Agency for Data Supply and Infrastructure (SDFI), the Danish EPA (MST), the Danish Coastal
Authority (KDI) and the Danish Environmental Portal (DMP). Special thanks are due to Anker Lajer Højberg for providing
the shapefile of ID15 catchments, and to Mark F. T. Hansen for verifying the connection between the ID15 catchments and
DKM.

Appendix A





Figure A1. Distribution of catchment attributes.



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
