# Peer review of "Consolidating a Physically Based Hydrological Model with Long Short-term Memory Networks"

_Hydrology and Earth System Sciences, 2023_

## Author Response (AR1)

**Response to Review #1 comments concerning HESS submission:**

**A National Scale Hybrid Model for Enhanced Streamflow Estimation - Consolidating a Physically Based Hydrological Model with Long Short-term Memory Networks**

Jun Liu, Julian Koch, Simon Stisen, Lars Troldborg, Raphael J. M. Schneider

Department of hydrology, Geological Survey of Denmark and Greenland, Copenhagen, 1350, Denmark

*Correspondence to*: Jun Liu (juliu@geus.dk)

The authors compare the performance of different types of hybrid models based on LSTM networks and a physically-based model, DKM, in estimating streamflow for Denmark. Generally, the hybrid models outperform LSTM rainfall-runoff model (LSTM-rr) in ungauged basins. They find the hybrid dynamic inputs LSTM model (LSTM-q) and the LSTM residual error model (LSTM-qr) have the overall best performance. The hybrid models also improve streamflow estimates in groundwater dependent basins. The study is interesting, and the authors provide a comprehensive discussion. My concerns on the study are as follows.

Reply: we thank the referee of reading the manuscript and providing helpful comments and suggestions. Here we reply to the comments point-by-point. The font color of the original comments from the reviwer is blue, while the font color of our replies is black, and the font color of the changes to the manuscript is red. We hope that these changes satisfy the requirements for proceeding with the publication of the updated manuscript.

In this study, LSTM-q and LSTM-qr are the optimal streamflow models for Denmark. Will the models be optimal in other regions in the world? I am worried that this might be a local conclusion.

Reply: We hold the opinion that LSTM models, particularly those utilizing simulations from physically based hydrological models, exhibit better performance also in other regions. We discussed this point in the manuscript and referenced previous studies (see lines 65-71 and lines 509-511). Although the accuracy of hydrological models varies from region to region, we expect that their simulations will contribute to improving the performance of hybrid LSTM models when being benchmarked against hydrological simulations and LSTM rainfall-runoff models Information from physically based models will and should always provide additional insights to deep learning models.

Changes: we added a sentence in Lines 514-515, which is as follows:

*Our results further conformed that LSTM models can be further enhanced by providing information from physical hydrological models.*

The authors discuss it in the Discussion section (Line 504-512) by comparing their conclusion with that from Konapala et al., (2020). They argue that the different best hybrid models in the two studies is due to the higher accuracy of DKM. Is it possible that the difference is also due to the different study domains, i.e., Denmark in this study and CONUS in Konapala et al., (2020)? In general, CONUS has a much deeper groundwater table depth than Denmark. The authors may

evaluate the performance of different hybrid models in various groundwater table depth ranges to check whether the conclusion will be changed.

Reply: The performance of LSTM-q and LSTM-qr is comparable with LSTM-qr being slightly better. The inputs of LSTM-q and LSTM-qr are the same, the only difference is the target variables. The target variable of LSTM-q is discharge and the target variable of LSTM-qr is the residual between simulated discharge and observed discharge. As stated in Konapala et al., (2020), LSTM-q is more correlated to the rainfall-runoff LSTM (figure 1(c)), while LSTM-qr is more correlated to the PBM. Our conclusion is that LSTM-qr outperforms LSTM-q because DKM is better than the PBM applied by Konapala et al., (2020).

We showed that the correlation between model performance with basin attributes (including groundwater table depth) and the correlation patterns of LSTM-qr and LSTM-q are very similar. We argue that the different conclusions of which hybrid setup is best between our study and Konapala et al., (2020) mainly relates to the diverging physical based model performances. It should be noted that other study, for example, Cho and Kim, (2022) used a well-calibrated model WRF-Hydro (NSE = 0.72 and R = 0.88) to predict residuals and they share our conclusion that the residual model performs better. We will present this point more clearly in the revised manuscript and will therefore revisit our presentation and discussion of the results.

Changes: we added a sentence in Lines 523-525, which is as follows:

*Cho and Kim (2022) used a well-calibrated model WRF-Hydro (NSE = 0.72 and R = 0.88) to predict residuals and they share our conclusion that the residual model performs better.*

The common practice is to separate the data into training, validation and testing sets in the time order. Why do the authors choose 2011-2019 as the validation period and 1990-1999 as the testing period? Does the selected testing period have fewer human impacts on streamflow?

Reply: DKM model was calibrated in 2000-2010 and tested in 1990-1999, so we kept this routine to develop the LSTM models. DKM considers human activities during testing, training, and validation period, for example, water extractions and wastewater flow modelling. We will explain the reasons of how we decided period for testing, training and validation in the updated manuscript.

Changes: we added a sentence in Lines 289-290, which is as follows:

*The training and testing period are the same as DKM to ensure the comparability of LSTM models and DKM simulations.*

In addition, Section 3.2 compares the event performance of LSTM hybrid models during the validation period (2011-2019). At least 80% of study data used in the section are validation data, which have been observed by the hybrid models during the hyperparameter tuning. The performance of the hybrid models is expected to be good, but may not deliver reliable information.

Reply: In section 3.2, the model was retrained with data from all stations and the training period is 1990-2010. We did not use validation data during the training period. To make this point clear, we plan to add some sentences of the setting in the updated manuscript, see lines 427-430.

Changes: to make the statement clear, we modified sentences as follows in the revised manuscript, see line 424-425.

*We set the training period from 1990 to 2010 and evaluated model performance on specific extreme events during the latest decade.*

In Table 1, why does the hybrid models include as input phreatic depths at both 100 and 500 m resolutions?

Reply: We have two versions of the DKM model distinguished by different resolutions: 100m and 500m. The finer resolution provides more details and a more accurate representation of the phreatic depth. However, at basin scale the high-resolution spatial patterns of phreatic depth are not relevant since they will be averaged across the entire basin. Therefore, we will remove the 100m phreatic depth in the updated manuscript to avoid complexity and just stick to the 500 m.

Changes: The 100m phreatic depth in table 1 has been removed. Accordingly, the attribute also removed from the LSTM models and Figure 5.

Please improve the quality of the figures particularly Figure 8. If possible, please also increase the font size in the figures.

Reply: Figure 8 was compressed so the quality is low. We will provide the original figures in the future.

Changes: We provided a larger figure in the revised manuscript, please see Figure 9 in the revised manuscript.

**Specific comments:**

Line 129-130: Might change "hidden unit sizes" to "hidden neurons".

Reply: The phrase will be changed in the updated manuscript.

Changes: We have changed the phrase in the revised manuscript.

Section 2.5: "Table 3" in Line 314 and 330 should be "Table 4".

Reply: the errors will be corrected in the updated manuscript.

Changes: This error has been corrected in the revised manuscript.

Table 5: Maybe only write the best evaluation scores in bold for better visualization.

Reply: We marked the values with the best evaluation scores in bold.

Changes: We have modified Table 5. Only NSE was kept as the predominant criteria to evaluate model performance. The remaining metrics have been moved to appendix. See Table 5 and Appendix B in the revised manuscript.

Figure 8: The word "bias" seems not be a right word to use in the caption, which suggests the error. Please consider replacing it with the word like "difference".

Reply: We will change the words in the caption of Figure 8.

Changes: We have changed the words in the caption of Figure 8 in previous version which supposed to be Figure 9 in the revised version of the manuscript. Here is a copy of the modified caption.

*Figure 9: A comparison of simulated streamflow differences between DKM and two LSTM models (LSTM-q and LSTM-qr). The first row depicts DKM simulated streamflow during high flow, low flow, and median flow conditions, the second row shows the differences between DKM simulations and the LSTM-q predictions, and the third row shows the differences between DKM and LSTM-qr. The percent diff in the figure is defined as the differences between LSTM model and the DKM, percent diff = $(Q\_LSTM - Q\_DKM) \times 100$.*

**Response to Review #2 comments concerning HESS submission:**

**A National Scale Hybrid Model for Enhanced Streamflow Estimation - Consolidating a Physically Based Hydrological Model with Long Short-term Memory Networks**

Jun Liu, Julian Koch, Simon Stisen, Lars Troldborg, Raphael J. M. Schneider

Department of hydrology, Geological Survey of Denmark and Greenland, Copenhagen, 1350, Denmark

*Correspondence to*: Jun Liu (juliu@geus.dk)

This manuscript proposes the use of different combinations of long short-term memory (LSTM) with a physical model-Danish Water Resources Model (DKM) to improve the accuracy of streamflow prediction. It is suggested that the hybrid model improved the model accuracy in ungauged and gauged basins. The authors further pointed out that the hybrid models could enhance the accuracy of extreme events. The knowledge gap is convincing, and the paper is clear. However, I have some comments that should be addressed before publication.

Reply: We thank the referee for his/her time and effort in reviewing our manuscript. We have revised the manuscript and responded to the comments point-by-point accordingly. The font color of the original comments is blue, while the font color of our replies is automatically black.

**Specific comments:**

1) In the introduction, the authors claimed that the DKM model is a well-established groundwater modeling system (Line 100). However, Nevertheless, this paper lacks observational evidence, particularly in results to support this claim. References are suggested.

Reply: We cited the evaluation reports of the DKM in lines 41-45. Detailed evaluation reports of DKM performance can be found in Henriksen et al., 2021 and Stisen et al., 2020, which also has been cited in the manuscript. We will also provide more publications of applications based on the DKM in the updated version of the manuscript.

Changes: we added three more references to the end of the sentence, which are relevant to the use of DK-model, see lines 103. Koch et al. (2021) modelled high resolution shallow groundwater with DK-model simulations, Schneider et al., (2022b) used random forest to downscale model results with coarse resolution to a higher resolution, and Henriksen et al., (2023) used DKM as a digital twin for the analysis of climate change adaption, water management. They all show the benefits of the DKM for research activities.

2) Sections 2.1 and 2.2.4 are overlapped and can be merged into a whole section.

Reply: We will modify the structure of section 2 and move section 2.1 to section 2.2.4.

Changes: We moved section 2.1 to section 2.2.4.

3) A table is suggested to compare the inputs and outputs of each type of model in section 2.3.

Reply: We will provide a figure to show the inputs and outputs of each type of LSTM model in the updated version of the manuscript.

Changes: we created a figure to show the inputs and outputs of different models, see figure 2 in the revised manuscript.

4) The caption of Figure 2 is not clear. Please explain each of the subplots.

Reply: The caption of Figure 2 will be modified in the updated manuscript.

Changes: we modified the caption for the figure. In the revised manuscript, we added a figure before, so Figure 2 becomes Figure3, and the modified caption is as follows:

*Figure 3: Daily time series of streamflow residuals (a) and error factors (c) between DKM simulated streamflow and observations at a hydrological station (ID: 51350461), the grey area shows observed streamflow time series. Autocorrelation of the time series are displayed in (b) and (d) to test white noise of residuals and error factors. The horizontal grey lines in (b) and (d) correspond to 95% (dash) and 99% (solid) confidence bands.*

5) In section 3.2, the operational forecasting framework uses only the observations of meteorological factors as model inputs. However, many studies try to combine the historical simulations or observations as model inputs which contributes to model forecasting. In the authors' cases, please comment on the impacts if considering the history series as model inputs.

Reply: In section 3.2, we evaluated the performance of LSTM-q and LSTM-qr in the operational forecasting framework, due to their higher performance. The inputs are climate forecasting and DKM simulations. Our plan for the operational forecasting framework is to use climate forecasting and DKM simulations as inputs to get a higher accuracy of streamflow. We are aware that some studies have promoted methods involving the use of LSTM with data integration to enhance streamflow forecasting (Feng et al. 2020). However, this topic is beyond the scope of the current manuscript, and it is part of our future work. We plan to include sentences discussing how we incorporate newly available observations into the operational framework in the updated manuscript, for example, after line 530.

Change: we mentioned data assimilation to improve the operational forecasting, see line 546.

6) In the discussion part, it is interesting to get the conclusion of model performance: LSTM-qr ≈ LSTM-q > LSTM-rr > LSTM-qf > LSTM-pf ≈ DKM. Could authors explain why the LSTM-rr performed better than the LSTM-pf, as the LSTM-pf is the pretraining and finetuning LSTM-rr?

Reply: Regarding the LSTM-pf model, it was pretrained based on all ID15 catchments, totalling 2830. The input data consist of climate forcing, and the target variable is DKM simulated discharge. The fine-tuning process took place at 276 gauged basins, employing the same climate forcings, but with the target variable being observed discharge. We assume the main reason for the poorer performance of the LSTM-pf model being primarily attributed to differences in target data between the pretraining (target: DKM simulated discharge) and fine-tuning phases (target: observed discharge). The DKM model has been calibrated in gauged basins, but its performance in ungauged basins remains unknown. Notably, basins with larger areas exhibit higher accuracy than smaller basins, but the majority of ID15 catchments is small basins and the representation of DKM simulations of real world is low. Despite using numerous basins for pretraining, significant data discrepancies persist between the pretraining and fine-tuning datasets. It can be hypothesized that the imbalance of basins used for pretraining and finetuning (2830 sim vs. 276 obs) is the reason why the LSTM pf model performs poorer

than the LSTM rr model. Future work should address such trade-offs and the role of epochs in both training rounds, which will certainly affect the overall result. We will extend the discussion of these results in the revised manuscript.

Change: we have some discussion in the revised manuscript, see line 525-537.

**Response to Review #3 comments concerning HESS submission:**

**A National Scale Hybrid Model for Enhanced Streamflow Estimation - Consolidating a Physically Based Hydrological Model with Long Short-term Memory Networks**

Jun Liu, Julian Koch, Simon Stisen, Lars Troldborg, Raphael J. M. Schneider

Department of hydrology, Geological Survey of Denmark and Greenland, Copenhagen, 1350, Denmark

*Correspondence to*: Jun Liu (juliu@geus.dk)

We have prepared a plan to address each point systematically, and the manuscript will be updated accordingly in the future. The original comments are highlighted in blue, while our responses are in black. We hope the responses fulfil the requested changes required to proceed with the publication of the updated manuscript.

The paper aims to investigate the advantages of utilizing a distributed Process-based Model (PBM) in implementing an LSTM representation for streamflow prediction. The researchers tested various traditional combinations to analyze the pros and cons of each configuration. They concluded that LSTM with the output of the PBM as input (LSTM-q) and an LSTM model learning the residual error of PBM (LSTM-qr) were the best models. One of the interesting findings of the study is that the hybrid model requires less memory (sequence length) than a simple LSTM (LSTM-rr). This indicates that by using PBM in LSTM, it can incorporate longer temporal dependences, which mitigates one of the issues of LSTM representation. Another notable finding related to this is that LSTM decreases performance in groundwater-dominated catchments, as suggested by other studies.

Reply: We appreciate the reviewer for reading our manuscript and providing valuable comments. We have planned to revise the manuscript based on their comments.

However, I have some major comments about the lack of clarity in defining the criteria for the best model and the explanation of some figures. Although the authors defined several metrics to evaluate the model, it was not clear which one or what combination of them was used to define the best model. Additionally, in many cases, the differences in performance are so small that they are probably not statistically significant.

Reply: We agree that the involvement of a group of metrics for the evaluation complicates the conclusions to be drawn from the results. We will use NSE as the basic index to evaluate the performance of different models. NSE is a comprehensive metric to measure overall fit of a model and how well the model captures the variability of observations. We will provide the results of NSE in the main part of the manuscript and provide the rest of the metrics in the Appendix. According to NSE, LSTM-q and LSTM-qr are comparable to each other, but better than the others (see table 1.). In case the differences in performance are not statistically significant, we consider to carry out a two sample Kolmogorov-Smirnov test https://docs.scipy.org/doc/scipy/reference/generated/scipy.stats.ks_2samp.html, to see if the metric distributions of two models are significantly different or not.

Changes: we decided to use the mean NSE based on the spatiotemporal split experiment to find the best model and rank the other models. We have modified the tables and figures as suggested by the reviewer and added explanations to all of

the figures showed in the manuscript. We admitted that the differences of performance are small when we do the hyperparameter tunning, which can explain that many studies used default hyperparameters to train LSTM models, and they got satisfactory results.

Moreover, some figures were presented without any further explanation. For instance, Figure 3 shows 16 subplots, but the text only mentions two lines about it. It's crucial to present figures that support the story presented in the paper, and if a figure isn't explained, it should go to the appendix. However, the authors should try to analyze each figure as much as possible because they will find more details that support their findings.

Reply: We will retain the crucial subplots (NSE) in the figure and move the others (KGE, NSE2, NSE_log, FHV, etc.) to the appendix. Overall, we will make sure to address each presented figure with sufficient detail in the revised manuscript.

Changes: we have modified most of the figures in the revised manuscript.

Minor comments.

Line 29: If you mention extra/interpolation, please explain why it is important for your goal.

Reply: We are going to add one more sentence after this line for the explanation. The potential sentence could be as follows:

Change: We have added some examples of model extra/interpolation, see lines 29-30.

*'such as supplementing the missing streamflow at stations, transferring the parameters to basins showing high hydrological similarities and predicting streamflow under future conditions.*

Line 59-60: The statement could mislead readers to believe that only DL methods experience a decline in performance. Please modify it.

Reply: We agree that the sentences are not rigorous. We will modify the statement as follows, some references will also by cited.

Changes: We have rewritten the statement. Please see lines 61-62 in the revised manuscript. Here is a copy of the sentence.

*While these models often demonstrate higher performance, accuracy may decrease when attempting to transfer them from gauged basins to ungauged ones, which is a common concern in the context of physical models as well (Winsemius et al., 2009; Ma et al., 2021).*

Line 97: Please use a software or a method to verify references as I found an incorrect citation. It should be De la Fuente et al. (https://doi.org/10.5194/hess-2023-252) instead of Fuente et al.

Reply: We will thoroughly review the references.

Changes: we have carefully checked the references.

Line 98: I agree with the sentence, but you should provide references, considering the "limited attention" given to the topic.

Reply: The references, initially placed in the middle of the sentence, will be moved to the end. Additionally, we will explore more recent references to ascertain their potential to support our statement.

Changes: we have added the references to the end of the sentence, see line 101-102. The modified sentence in the revised manuscript is as follows.

*While previous studies have explored the effects of snow melting on LSTM modelling, limited attention has been given to the impacts of groundwater variations on LSTM rainfall-runoff modelling (Frame et al., 2021b; Fuente et al., 2023; Kratzert et al., 2019a; Wang et al., 2022).*

Table 1: It would be very useful to add some summary statistics, such as range and mean.

Reply: This is a nice suggestion; we will include the ranges and means of these attributes.

Changes: we have modified the table, statistics, including minimum, maximum, median, mean, and standard deviation, have been added, see table 1 in the revised manuscript.

Line 291. Why did you change the loss function? Different loss functions emphasize different components of the error.

Reply: NSE is a comprehensive metric for measuring the differences between simulated discharge and observed discharge. However, we don't consider it as a suitable metric for residuals or error factors, due to their very different timeseries characteristics you see in Figure 2.  This is why we use RMSE in the loss function for residual and error factor models.

Changes: we retrained the models and the loss function we used is RMSE. Please be aware that the use of RMSE gives slightly different performance of LSTM-rr and LSTM-q.

Line 307. Line 307: The hyperparameter search could generate some inconsistencies because only 6.2% of the parameter space is being explored (100/1620). To address this, it is recommended to fix hyperparameters with low sensitivity such as LR, BC, NE, and DR. This way, a more detailed exploration of the hyperparameters that matter can be carried out.

Reply: We defined the candidate values because we didn't have a good sense of the optimal values for the hyperparameters. We selected the values based on previous studies. However, we agree with this suggestion; some of the hyperparameters are less sensitive according to our results. This is a limitation of our study, and we will discuss it in the updated manuscript.

To facilitate a discussion on the relationship between model performance and sensitive hyperparameters (i.e., Figure 9), we intend to conduct an additional hyperparameter search. We will use a limited set of values for the number of epochs (20, 25, 30, 35) and hidden unit size (64, 128, 256), while fixing the dropout rate (0.3), learning rate (10^-3), batch size (128). and the sequence lengths (10, 30, 60, 90, 180, 270, 365, 730). The total number of hyperparameter combinations will be 96 (4*3*8) for each model. Subsequently, we will reproduce Figure 9.

Changes: we used the above hyperparameter settings, updated all the results with the optimal hyperparameters, reproduced and the figures with new results (including Figure 9). Please see the updated figures in the revised manuscript.

Table 3. How much is the difference with the second best? It is a little strange that models with the simulated streamflow as input have more hidden cells than the baseline that is learning the entire dynamic of the system.

Reply: We will provide the range of NSE, including minimum, maximum, median, and mean values of the results in table 3. The models with more hidden cells not only simulate streamflow but also incorporate average water content in soil layers, actual evapotranspiration, and depth to the phreatic layer as dynamic inputs. Therefore, the LSTM model could be more complex and have more hidden cells.

Changes: Models with different hyperparameters show similar performance, see the standard deviation (std) in table 3. So, the second-best model is very close to the best one. This indicate that impacts of hyperparameters are insignificant, many studies therefore used default values, which are reasonable. The hidden size is 64 for all the models after we retained the models except LSTM-qr has a hidden size of 128.

Table 5. If the difference between the mean and median is not mentioned, it's recommended to either delete one of them or move the discussion to the appendix.

Reply: We will retain the mean NSE values in the updated manuscript and relocate the other metrics to the Appendix.

Change: we have modified Table 5, which only shows the statistic of NSE, the results of hyperparameters have been moved to appendix.

Line 348. LSTM-q outperforms LSTM-qr only for NSE2. Please, check your analyses.

Reply: We will revise the analysis of the sentence.

Change: We have a new set of hyperparameters, so the sentences have been modified, see line 330-333.

Line 353. In the text, it is mentioned that LSTM-rr has a lower NSE_log than DKM, but the table shows the opposite.

Reply: The argument behind this statement relies on the mean values of NSElog. The mean NSElog of LSTM-rr is 0.36, whereas the mean NSElog of DKM is 0.41 in the spatiotemporal split experiment. However, the median NSElog of the two models is contradictory, causing confusion in the statement. To address this, we plan to simplify Table 5 by including only the mean values of NSE, while the remaining metrics will be moved to the appendix. We will also rephrase the sentence in line 353.

Change: we have a new set of hyperparameters, so the results were changed. Please see line 330-336 and Table 4.

Figure 3. You did not analyze the figure. Therefore, you should either delete or move it to the appendix. However, I believe that this figure is more informative than Table 5, so I encourage you to describe it.

Reply: We will keep NSE as the main metric in the revised manuscript, so Figure 3 and Table 5 will be kept short to only show the results of NSE. The results of the other metrics will be moved to the Appendix in the updated manuscript.

Change: Figure 3 has been modified, only the results of NSE are presented and the rest metrics have been moved to the appendix.

Figure 3. Cumulative distribution functions (CDFs) are typically displayed with metrics on the x-axis and probability on the y-axis. To improve clarity, it is suggested to limit the axis for KGE and NSE between [0,1]. This will provide a better visualization of the behavior of each line. Additionally, you may consider showing only some of the metrics in detail, while the rest could be placed in the appendix.

Reply: We will modify Figure 3 by following the suggestions and displaying the CDFs with metrics on the x-axis and probability on the y-axis. Considering that some of the models have negative KGE and NSE values, we initially limited the range to [-0.5, 1]. However, we will change these settings and restrict the range of NSE and KGE to [0, 1]. We will retain the first row of Figure 3, which only shows the results of NSE. The remaining rows of Figure 3 will be included in the appendix.

Change: We have plotted new subplots with metrics on the x-axis and probability on the y-axis, and limited the axis for NSE and KGE between [0, 1]. Only NSE are displayed in the modified figure, the rest metrics have been moved to the appendix.

Line 360. A value greater than zero is still unsatisfactory. Please rephrase this sentence.

Reply: We will modify this sentence. According to Moriasi et al. 2007, NSE>0.5 indicates model performance is satisfactory, we will count the number of basins with NSE higher than 0.5.

Change: The sentence has been rephrased and the modified sentence is as follows:

*DKM exhibits satisfactory performance (NSE > 0.5) in 72% of basins from the temporal split experiment and 64% from the spatiotemporal split experiment. There are seven stations from the temporal split experiment and five stations from the spatiotemporal split experiment that have negative NSE values.*

Line 363. It can be difficult for someone from another country to identify specific areas. Adding a latitude-longitude grid can help with this issue.

Reply: We have the names mentioned in the manuscript in Figure 1(c), which may not be clear to see. To enhance clarity, we will add a latitude-longitude grid to Figure 1(c) and change the font size of the names.

Changes: We recreated Figure 1 to show the location of Denmark with latitude-longitude grid. A map of Denmark with names of geographical regions is also shown in the figure. Please see Figure 1 in the modified manuscript.

Figure 4. If you are not going to describe the other maps, move them to the appendix and present only the histogram. The current colour scheme makes it difficult to identify patterns. To address this, Figure a) could benefit from a traffic light palette (green for good, yellow for regular, and red for bad). Meanwhile, the other figures could use a palette that transitions from white in the center to either red or blue on the extremes, respectively. This approach will allow readers to better focus on relevant changes.

Reply: We will modify the figure accordingly. Our plan is to retain the results of DKM, LSTM-q, and LSTM-qr in the figure while relocating the others to the Appendix. We will also adjust the colors of the legend as suggested.

Changes: We have modified Figure 4 suggested by the reviewer. NSE of Fig 4a is shown in a palette with four colours: very good (0.75 < NSE < 1.00) in blue, good (0.65 < NSE < 0.75) in orange, satisfactory (0.50 < NSE < 0.65) in red, and unsatisfactory in (NSE < 0.50) green. For the ΔNSE maps, we keep in the subplots because they are described in the manuscript, the palettes have a centre value of 0 and coloured white, extreme high values are in red and extremely low values are in blue. Please see the modifications in the revised version.

Figure 5. It is difficult to distinguish between the colours of LSTM-rr and LSTM-qf. Adding the KGE or NSE of each model to the legend would provide an additional comparison.

Reply: We will change the colour of LSTM-qf, for example, to brown. NSE will be appended to the end of the model name in the legend.

Changes: we have changed the colours of LSTM-rr and LSTM-qf, and we added the NSE to the legend for better comparison. Please see the Fig.7 in the modified manuscript.

Line 392. Please provide references to support the fact that this finding is not surprising. As far as I know, lumped models tend to perform poorly in predicting larger areas when there is a non-uniform distribution of precipitation. However, larger catchments are usually influenced more by baseflow which should be easier to predict. Hence, to validate this conclusion, it would be helpful to compare it with other studies conducted in the region.

Reply: The DKM model performance report documents higher accuracy over large basins. We will provide the references in the revised manuscript. We also need to mention that most of our catchments are still quite small compared with larger rivers globally, and the spatial variations of precipitation is not significant and spatial aggregation is not that important.

Changes: we have added the DKM performance report the end of the sentence, see line 395.

Line 397. To gain a better understanding of the region, please add the range of groundwater levels.

Reply: We will provide the general ranges of groundwater depth in Table 1, and the corresponding sentence.

Change: we have added the range of groundwater levels in Table and mentioned the range of groundwater levels in the sentences.

Line 402. I would like to draw your attention to the fact that the correlation analysis was conducted using only one variable. This means that any interaction between attributes has been overlooked. Additionally, the Pearson correlation only represents linear correlations, which underestimates more complex relationships. To address these issues, I suggest using the Spearman correlation and discarding low correlation values. Alternatively, you could build a random forest model to examine how the combination of attributes affects performance.

Reply: We will recalculate the correlation between model performance and basin attributes with Spearman correlation. However, we won't go deep to analysis the interaction between attributes and model performance. We agree that two or more variables coherently contribute to model performance, but that is not the main objective.

Change: We compared both Pearson and Spearman correlation. Spearman correlation is hard to use in this case, because many attribute values are very similar, not allowing for a reasonable ranking., though we can see a rising or falling trend from the scatter plot. Our purpose is to get a general sense of the relationship between basin attributes and NSE. So, we kept using a fitted linear relationship for the correlation.

Figure 6. Please double-check the color bar and ensure that the white color is set to zero.

Reply: We will modify this figure, changing the colour bar to ensure that white represents zero in the updated manuscript.

Change: We have modified Fig 6, we changed some of the attributes and recalculated the correlation, because of the retraining of the LSTM models. Please see the modified figure in the updated manuscript.

Line 428. The sentence "chosen based on their superior performance" is not accurate as the performance of the LSTM models is similar. Additionally, the models were not statistically compared.

Reply: We selected the two LSTM models based on the mean NSE (0.63) from the spatiotemporal split experiment. Their mean NSE is slightly higher than that of the other models.

Change: we have changed the criteria of ranking model performance. The best model was determined by the performance in the spatiotemporal split experiment according the mean NSE. After we retrained all the LSTM models with modified inputs, new hyperparameters and different loss functions. LSTM-q have a NSE of 0.80, with LSTM-qr have a NSE of 0.79. It is reasonable to assume that LSTM-q is the best model.

Figure 7. The figure contains too much information with the density function and histogram. It would be better to only show the density function and widen the figure.

Reply: We will remove the time series in the subplots and change the size of the figure to make it wide and fits the width of the paper.

Change: we have removed the results of LSTM-qr and the histogram, please see the modified results in Figure 9.

Line 445-446. Could you describe the locations of these regions?"

Reply: We will carefully check the names and locations in the manuscript and make sure they are properly shown in the study area figure (Figure 1).

Change: we have prepared a new figure of Figure 1, with the names of different regions. Please see the locations in the updated figure.

Line 448. Figure 7 exhibits an underestimation of the streamflow. Can you explain the source of this statement?

Reply: Figure 7 shows the streamflow for specific events. For instance, Figure 7a illustrates the average streamflow of all stations over a 30-day period starting from December 20, 2011. However, in line 448, we discuss the long-term changes in streamflow from 2011 to 2019.When compared with the DKM shown in Figure 8b, the LSTM model demonstrate lower values in the west of Denmark but positive values in the east for high flow (Figure 8b, e, h). LSTM-q exhibits better performance than DKM, resulting in simulations that are closer to the observations. Consequently, when DKM indicates higher discharge values, there is a higher probability of overestimation. The expression in this sentence is misleading, and we will rewrite it in the updated manuscript.

Changes: the analysis and statement of this sentence was unclear. We have rewritten the paragraph in the updated manuscript, please see lines 426-439.

Line 460. Could you please clarify on which results this conclusion is based?

Reply: we will rewrite the sentence and clarify the results:

Change: we have added the information of where the results this conclusion is based, see line: 469-472. The modified sentences are as follows:

*The results revealed that utilizing LSTM models, especially the hybrid schemes that were coupled with physically based simulations, exhibited better performance for both overall performances spanning a decade and specific hydrograph extreme events, see results in section 3.1 and section 3.2.*

Figure 9. Why aren't the values from Table 5 included in this figure? Also, why is LSTM-rr showing lower values than the others?

Reply: Figure 9 is based on the evaluation results of the spatiotemporal split experiment, so it should include the values from Table 5. We will carefully check Figure 9. LSTM-rr has lower performance in the spatiotemporal split experiment compared to other hybrid models, which is one of the conclusions of the manuscript stating that hybrid models have improved accuracy compared to benchmarks, i.e., LSTM-rr and DKM.

Changes, we have retrained all the models for optimal hyperparameter searching. Figure 9 in previous manuscript has been updated with new candidates of hyperparameters, see Figure 11 in the updated manuscript. The values from Table 3 and Table 4 are included in Figure 10.

Line 481. If there are not many studies that have made this comparison, you should include references to those studies.

Reply: We will add the references which used the spatiotemporal hold-out experiments in the updated manuscript.

Change: we have added references of using spatiotemporal hold-out experiments for model performance evaluations, see line 490.

Line 498. Can we attribute this improvement to the use of longer memory, as shown in Figure 9?

Reply: We can attribute the improvement of LSTM-rr to longer memory, which requires a longer sequence length to achieve better performance, as shown in Figure 9. However, the improvement in the hybrid model is attributed to the provision of more information about complex hydrological processes during the training of LSTM models. Hybrid LSTM models do not show an improvement with longer input sequences in Figure 9.

Changes: we have modified Figure 9. Longer memory improved the performance of LSTM-rr, but it is not significant for the hybrid models, as the increasing of sequency length, the performance of LSTM-q and LSTM-qr is not significant. The performance of LSTM-qf gets event worse when the sequence length is getting longer than 180 days.

Line 500. You must explain your ranking process as you are using multiple metrics. Are you using one of them, their average, or some combination? Additionally, many of the models may not be statistically significant.

Reply: We plan to use mean NSE as the main metric to rank the models. According to the mean NSE values in Table 5, the order is LSTM-qr ≈ LSTM-q > LSTM-qf > LSTM-rr > LSTM-pf = DKM for the spatiotemporal split experiment. Additionally, the order is LSTM-qr > LSTM-q = LSTM-rr > LSTM-qf > LSTM-pf > DKM for the spatiotemporal split experiment. Considering that the performance of LSTM-q is higher than LSTM-rr in the spatiotemporal split experiment, we have concluded that LSTM-qr and LSTM-q are the two best models.

Changes: we used the NSE from spatiotemporal split experiments to rank the models, see line 510. We agree that the performance of the best and the second model are statistically insignificant. However, if we look at the performance in Fig.10. LSTM-q have an overall performance, followed by LSTM-qr, LSTM-qf and LSTM-rr.

Line 525. Could you please clarify what you mean by shallow structures? It seems contradictory to say that 256 hidden cells in parallel with 365 days of sequence length is a shallow structure. Many studies have shown that using more than 2 layers in a series does not result in significant improvement. This suggests that a single layer has sufficient complexity to capture the necessary processes.

Reply: The term 'shallow structure' here implies that we only trained one type of LSTM model encompassed the entire study area and its complex hydrological processes. In contrast to the approach of training distinct LSTM models for each sub-region, a single LSTM model lacks versatility. We acknowledge that the phrase may be misleading, and as a result, we will remove the term 'shallow structure' from the sentence.

Changes: we have rewritten the statement as follows, see line 539-541, and as copy as follows:

*However, the LSTM networks presented in this study were trained using a limited number of gauged basins, potentially failing to encompass the full spectrum of hydrological regimes, which decreased their capacity to capture certain features effectively.*

Line 526. I do not believe that the features and attributes are deeply hidden. The problem lies in the representation and inputs used to extract information.

Reply: We will modify the sentence.

Changes: we have modified the sentence, and the revised version see line 539-541.

Line 529-530. Instead of making your representation deeper to increase its complexity, you should try using a multi-representation approach. Different representations or architectures can capture different pieces of information. Using local models may alleviate the issue, but it does not solve it. This is like trying to approximate a high-degree polynomial by using only order 2 polynomial segments. Adding more order 2 polynomials segments does not increase the complexity; it only segments the extraction of information.

Reply: we are not very sure about the multi-representation approach. It will be great if the reviewer can explain more about it on our case Subsequently, we will make the necessary modifications to our manuscript.

Change: we have rewritten the statement of the sentence in line 546-548, and a copy as follows:

*Enhancing the neural networks with a multi-representation approach, or developing specific DL models for different regions distinguished by regime information could be alternative solutions in the future (Hashemi et al., 2022).*

Line 542. Some researchers have used graph neural networks with or without routing parameters in training. Mention them.

Reply: This comment is very informative and expands our knowledge in this area. We will explore graph neural networks and cite them properly here.

Changes: we have added references of GNN to line 564.

Line 559. Are these values significantly different from the LSTM-rr? for this reason, you must be more specific about what was your final multi-objective criteria.

Reply: We plan to use NSE to rank different models. The remaining indices will be included in the appendix for discussion.

Changes: we used the mean NSE calculated in the spatiotemporal split experiment as the only metric to rank the models. LSTM-q has a mean NSE of 0.64, and LSTM-rr has a mean NSE of 0.60. This difference indicates that they are significantly different.

Line 561-562. Why were LSTM-qr and LSTM-q not able to beat always to the DKM model despite using its outputs?

Reply: There are two reasons the LSTM model failed to improve streamflow estimation:

(1) The DKM model has been calibrated and demonstrates good performance in some basins (NSE > 0.8), indicating limited room for further improvement by the LSTM.

(2) We trained a single LSTM for all basins, integrating all inputs spatially, leading to the omission of some information. So, despite the complex internal structures of LSTM models, they remain catchment-lumped models, whereas the DKM is a fully distributed model.

Changes: We have discussed the reasons in Lines 541-546 in the revised manuscript, here is a copy:

*The catchments have a large variety of static attributes spatially, and the hydrological regimes change significantly across Denmark. While the hybrid schemes offer enhanced information and mitigate the issue of limited input data, such as LSTM-q and LSTM-qr, they fall short in distinguishing stations requiring further improvement or those already meeting requirements from the physical model. Consequently, this deficiency may explain why LSTM models exhibit inferior performance at few stations when compared to DKM. Enhancing the complexity of the neural networks, or developing specific DL models for different regions distinguished by regime information could be alternative solutions in the future (Hashemi et al., 2022).*

Line 567-568. Do you have any suggestions or ideas?

Reply: We will propose suggestions to potentially enhance the prediction of extreme events. For instance, we can employ objective functions that emphasize extreme values, such as $NSE^2$, to train the LSTM model.

Changes: we added the following sentence in the end to give some suggestions. See line 585, and the copy as follows:

*Future considerations may include employing alternative objective functions like $NSE^2$ or manually augmenting the occurrence of peak flow during model training.*

Line 570. It would be helpful if you could mention the complex hydrological processes.

Reply: We will rewrite the sentence and mention the complex hydrological processes in the updated version.

Changes: The modified sentence is in line 587-588, and a copy as follows:

*Previous studies concentrated on gauged basins, whereas this study makes contributions to the topic through a national-scale analysis, highlighting areas characterized by complex hydrological processes.*

---

## Author Response (AR2)

**Response to report #1 comments concerning HESS submission:**

**A National Scale Hybrid Model for Enhanced Streamflow Estimation - Consolidating a Physically Based Hydrological Model with Long Short-term Memory Networks**

Jun Liu, Julian Koch, Simon Stisen, Lars Troldborg, Raphael J. M. Schneider

Department of Hydrology, Geological Survey of Denmark and Greenland, Copenhagen, 1350, Denmark

*Correspondence to*: Jun Liu (juliu@geus.dk)

Thanks for authors'efforts on replying the comments and making revisions. My few comments on the study are as follows.
Reply: We thank the referee for reading the manuscript again and providing helpful comments and suggestions. Here we reply to the remaining comments point-by-point. The font colour of the original comments from the reviewer is blue, while the font colour of our replies is black. We hope that these changes satisfy the requirements for proceeding with the publication of the updated manuscript.

1) In figure 6, authors compared the predictions from DKM, LSTM-rr and LSTM-q in A and B basins. Authors pointed that the different performances of LSTM-rr and LSTM-q in two basins were caused by the basin attributes. However, I have a doubt because the LSTM-q model considered the additional DKM simulations as inputs compared to LSTM-rr model, and both models have taken the basin DEM and slope as static inputs. The DKM got a good performance in base flow prediction in basin B, so I wonder if the poor performance of LSTM-q model was caused by nonoptimal model optimization.
Reply: The two stations shown in Figure 6 were not used for training, they are two stations from spatiotemporal split-sample experiments. The estimation of discharge at such two stations is estimated from their inputs and attributes. If the inputs and attributes are extreme cases among the training dataset, the LSTM prediction may give a low accuracy.
We agree with the argument that LSTM-q is nonoptimal. An optimal LSTM-q could figure out how to trust DKM simulated discharge. For example, ideally, the LSTM model follows DKM simulated discharge in the basins where DKM has good performance, but LSTM simulates discharge more based on precipitation and other variables in basins where DKM has poor performance. However, it seems the trained LSTM-q model cannot figure it out, and the simulations are poorer compared with DKM simulations in a few basins. There are many reasons (and we don't think it solely can be linked to model optimization), such as the inputs are insufficient and the representativeness is low, the model design is imperfect. We have discussed this point in lines: 539-545.

2) In figure 7, I am curious of the correlations between different basin attributes and models. If authors select the positive correlated parameters as model inputs instead of all attributes, does it improve model predictive performance?.
Reply: Figure 7 shows linear correlation between catchment attributes and model performance. A negative value indicates the model simulates streamflow less accurately with rising attribute value, but the attribute likely remains important. For example, with deeper shallow groundwater level, the models generally struggle to accurately simulate streamflow.

Despite some low correlations in this simple linear correlation analysis, the respective covariates still might have informational value for the more complex LSTM model. Also, the hyperparameter search allowed for complex enough architectures to be able to ingest a row of static attributes, not having to limit ourselves to very few.

3) In figure 8 caption, there is no LSTM-qr shown in the figure?

Reply: We did not show the results of LSTM-qr, we have corrected the caption of figure 8, and we removed 'LSTM-qr' from the caption.